# Improving Sampling for Masked Diffusion Models via Information Gain

Kaisen Yang [1]   Jayden Teoh [2]   Kaicheng Yang [3]   Yitong Zhang [4]   Alex Lamb [5]

## Abstract

Masked Diffusion Models (MDMs) enable flexible decoding orders, yet existing samplers remain largely greedy, selecting locally certain tokens without accounting for their downstream effects. We show that this myopia can increase cumulative uncertainty and lead to suboptimal generation. To address this, we propose the **Info-Gain Sampler**, a training-free decoding method that uses the bidirectional structure of MDMs to balance immediate uncertainty with the information gained over remaining masked positions. Across reasoning, coding, creative writing, and image generation tasks, Info-Gain Sampler consistently outperforms existing MDM samplers, improving average reasoning accuracy by 2.9–11.6 percentage points and achieving a 62.8% average win rate in creative writing. The code is available at https://github.com/yks23/Information-Gain-Sampler.

## 1. Introduction

Masked Diffusion Models (MDMs) have emerged as a powerful alternative to the dominant autoregressive paradigm for discrete sequence generation (Austin et al., 2021a; Lou et al., 2024; Nie et al., 2025). By leveraging bidirectional attention, MDMs break free from strict left-to-right generation, granting unprecedented flexibility in decoding paths (Rombach et al., 2022). This flexibility unlocks superior performance in tasks requiring bidirectional attention, such as code infilling, biological sequence design, and long-horizon planning tasks (Ye et al., 2025; Gong et al., 2025; Ye et al., 2024; Wei et al., 2026).

However, this potential remains largely untapped due to a

training–inference mismatch. While MDMs are trained under random masking patterns, inference entails a multi-step, order-sensitive decoding process. Navigating the large space of possible decoding orders therefore requires a sampler that carefully selects which tokens to reveal next. Consequently, generation quality is heavily dependent on the effectiveness of the sampler (Kim et al., 2025a).

Existing samplers predominantly rely on **local certainty heuristics** such as confidence to greedily select the next decoding target (Chang et al., 2022; Ye et al., 2025; Huang et al., 2025; Kim et al., 2025a). In Section 3.1, we argue and demonstrate that such samplers are often nonrobust due to the myopia of local heuristics: they ignore the long-term impact of current decoding decisions on future uncertainty. Consequently, they frequently prioritize tokens that appear syntactically *confident* but are semantically suboptimal, leading to error propagation and compromised generation quality.

In contrast to autoregressive models (ARMs), where the causal nature makes evaluating the downstream effect of a token choice computationally prohibitive, the bidirectional nature of MDMs offers a distinct advantage: it enables us to assess how a token decoding decision influences the uncertainty across all remaining masked positions immediately.

Leveraging these insights, we propose the **Information Gain (Info-Gain) Sampler**, a decoding framework that departs from greedy certainty-based sampling. Instead of merely refining the certainty scoring function, the Info-Gain Sampler additionally evaluates decoding actions by how much they reduce the uncertainty in remaining masked tokens. By balancing *immediate certainty* with *information gain*, our method prioritizes globally informative decisions and yields more robust decoding trajectories. Our contributions are threefold:

(1) We empirically identify the fundamental limitations of existing greedy certainty-based samplers in MDMs through failure case analyses.

(2) We introduce the Info-Gain Sampler, which balances immediate costs with future information gain via a simple yet effective objective. We also propose computationally efficient implementations of Info-Gain Sampler.

(3) We extensively evaluate Info-Gain Sampler against other

---

[1]Department of Computer Science and Technology, Tsinghua University [2]Massachusetts Institute of Technology [3]Shanghai Jiao Tong University [4]Beihang University [5]College of AI, Tsinghua University. Correspondence to: Alex Lamb <alex6200@gmail.com>.

*Proceedings of the 43rd International Conference on Machine Learning*, Seoul, South Korea. PMLR 306, 2026. Copyright 2026 by the author(s).

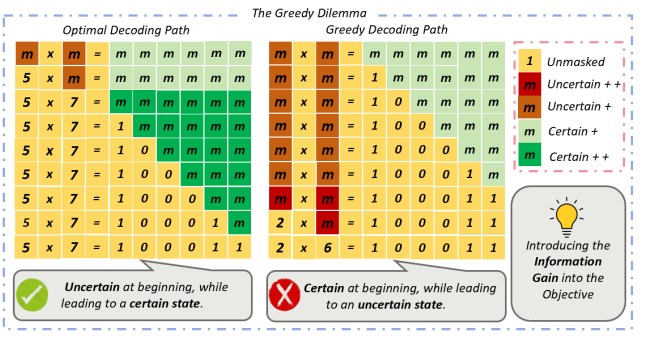

(a) The dilemma of greedy certainty-based sampling.

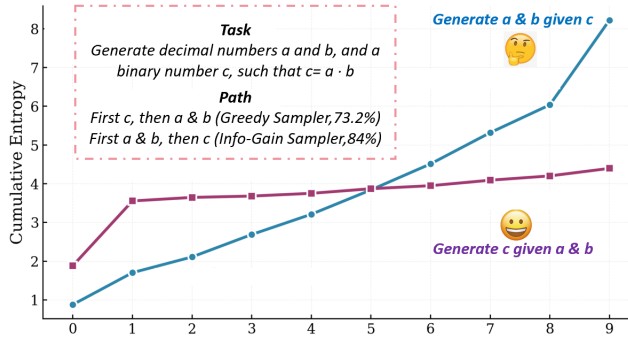

(b) Evolution of cumulative uncertainty.

*Figure 1.* **Motivation:** Analysis of decoding strategies on the one-way multiplication experiment. (a) Illustrates the contrast between the suboptimal path chosen by the greedy certainty-based sampler and the optimal path, motivating the introduction of the Info-Gain Sampler. (b) Shows the evolution of cumulative uncertainty throughout the decoding process. While the greedy sampler prioritizes decoding $c$ first (73.2%) due to its immediate high confidence, it leads to failure because of the task's one-way nature. In contrast, the Info-Gain Sampler optimizes global uncertainty by resolving high-entropy factors $a$ or $b$ first (84.0%), ensuring a successful decoding trajectory.

sampler baselines across diverse pretrained MDMs and benchmarks. The results show that Info-Gain consistently outperforms these baselines across math, coding, planning, writing, and image generation tasks.

## 2. Preliminary

### 2.1. Masked Diffusion Models (MDMs)

We consider discrete data over a vocabulary $\mathcal{V} = \{1, \ldots, V\}$ with sequence length $L$, and let $0$ represent the special mask token. A state is denoted as $z_t \in (\mathcal{V} \cup \{0\})^L$ for discrete time steps $t = 0, 1, \ldots, T$, where $z_t^\ell$ is the token at position $\ell$. The data distribution $p_{\text{data}}$ is defined over fully unmasked sequences in $\mathcal{V}^L$, corresponding to $z_0$.

**Forward process.** The forward process gradually corrupts a clean data point $z_0 \sim p_{\text{data}}$ by independently masking each coordinate over $T$ steps. At each step, tokens are progressively replaced by the mask token $0$ according to a fixed schedule, such that at time $T$, the state is fully masked, i.e., $z_T = (0, \ldots, 0)$.

**Reverse process and training.** To generate samples, we learn to reverse this forward process. A denoising network $p_\theta^\ell(\cdot|z_t)$ predicts, for each masked position $\ell$, the distribution of the original token $z_0^\ell$. The model is trained by minimizing the evidence lower bound, which reduces to a weighted cross-entropy loss over masked positions. As revealed in (Kim et al., 2025a), this loss weights all possible infilling problems equally, meaning the optimal $p_\theta$ learns to predict any masked token given any context.

**Sampling.** Sampling starts from the fully masked state $z_T = (0, \ldots, 0)$. At each step $t$ (going backwards from $T$ to 1), given current state $z_t$, the model computes distributions $\{p_\theta^\ell(\cdot|z_t)\}_{\ell \in \mathcal{M}_t}$ for all masked positions $\mathcal{M}_t := \{\ell \mid z_t^\ell =$

$0\}$ in a single forward pass.

A *sampler* $\pi$ determines how to use these distributions to produce the next state $z_{t-1}$. Unlike autoregressive models where the sampler decides the next token at a fixed position, a sampler for MDMs decides which masked positions to fill and what tokens to assign. Specifically, it selects a subset $A_t \subseteq \mathcal{M}_t$ to unmask and, for each $\ell \in A_t$, assigns a token $\hat{x}^\ell \sim p_\theta^\ell(\cdot|z_t)$ (optionally with temperature scaling or top-$k$ filtering). This yields an action $a_t := \{(\ell, \hat{x}^\ell) \mid \ell \in A_t\}$. Applying $a_t$ to $z_t$ produces $z_{t-1} = \text{APPLY}(z_t, a_t)$, where each selected position is filled and all others remain unchanged. This process repeats until reaching $z_0$, yielding a fully generated sequence $z_0 \in \mathcal{V}^L$.

**Certainty-based samplers.** A widely used family of samplers follows a two-stage procedure at each step. First, *token sampling*: for each masked position $\ell \in \mathcal{M}_t$, draw a candidate token $\hat{x}^\ell \sim p_\theta^\ell(\cdot|z_t)$. Second, *position selection*: select which positions to actually unmask using a *certainty score* $\phi(\ell, z_t)$ that measures the model's confidence at position $\ell$. Common choices for $\phi$ include the top-1 probability of the predictive distribution (confidence), the negative entropy of the predictive distribution, or the margin between the top-2 probabilities (Chang et al., 2022; Ye et al., 2025; Kim et al., 2025a). A formal description of mainstream certainty-based samplers is provided in Appendix A. Given a time-dependent budget $K_t$ (the number of positions to unmask at step $t$, determined by a predefined scheduling function), the sampler selects the subset $A_t^*$ that maximizes total certainty:

$$A_t^* = \underset{A_t \subseteq \mathcal{M}_t, |A_t| = K_t}{\operatorname{argmax}} \sum_{\ell \in A_t} \phi(\ell, z_t). \quad (1)$$

The final action is then $a_t^* = \{(\ell, \hat{x}^\ell) \mid \ell \in A_t^*\}$. This "most certain first" strategy fills positions where the model

is most confident, leaving harder decisions for later steps when more context is available.

# 3. Method

## 3.1. Motivation

Existing certainty-based sampling methods typically employ a greedy strategy. These methods aim to minimize error accumulation by prioritizing the most certain positions. While this approach effectively reduces immediate uncertainty and enhances short-term reliability, it essentially performs a greedy optimization of the cumulative uncertainty. This leads us to a fundamental question:

> **Question 1:** Is greedy optimization sufficient for minimizing cumulative uncertainty across steps?

To quantify the uncertainty throughout a generation process $\tau = z_T \rightarrow z_{T-1} \rightarrow \ldots \rightarrow z_0$, we first introduce **Cumulative Entropy** $\tilde{H}$ over $\tau$ as a key metric, defined as:

$$\tilde{H}(\tau) := \sum_{t=T}^{1} C(a_t \mid z_t), \qquad (2)$$

where $C(a_t \mid z_t) = \sum_{\ell \in A_t} H^{(\ell)}(z_t)$ represents the sum of marginal entropy for the tokens selected at step $t$, given by the model's output distribution. This metric quantifies the total uncertainty accumulated throughout the decoding trajectory.

To explore this question, we present two case studies that illustrate the limitations of greedy samplers.

**Case Study 1: One-way Multiplication.** The model is tasked with generating an equation $a \times b = c$, where $a$ and $b$ are decimal factors and $c$ is a binary product. This task is inherently one-way: computing a product from factors is straightforward, while factoring is not only computationally difficult but also particularly challenging for MDMs to generate, especially when the product is represented in binary format. Two decoding paths emerge: (i) *Product-first*, and (ii) *Factor-first*.

A greedy sampler mistakenly favors path (i) because the binary digits of $c$ exhibit lower per-token uncertainty (requiring a choice only from $\{0, 1\}$) than the decimal digits of $a$ and $b$ (which have 10 possible values). As shown in Figure 1, by minimizing immediate uncertainty, the greedy strategy commits to a product without fixing the factors, leading to incorrect equations and high residual uncertainty. Conversely, an optimal strategy would resolve the higher-uncertainty factors $a$ and $b$ first; once fixed, $c$ can be determined with nearly zero uncertainty. Empirically, the greedy certainty-based sampler (represented by Entropy (Ye et al., 2025)) prioritizes path (i) with 73.2% probability, leading to significantly higher cumulative uncertainty.

**Case Study 2: Binary Judgment.** In this experiment, the model is tasked with judging the truth of an arithmetic statement using the template: "[reasoning-masks] The answer is (Yes/No): [answer-mask]". The answer token typically exhibits lower local uncertainty as it is constrained to a binary choice (Yes/No), whereas the reasoning steps involve much higher uncertainty. Consequently, greedy samplers tend to decode the answer token prematurely, making a commitment before the underlying reasoning is resolved. This leads to incorrect judgments and leaves high residual uncertainty in the reasoning positions.

To analyze this, we compare greedy certainty-based samplers against an auto-regressive (AR) baseline, which naturally resolves high-uncertainty reasoning before the binary answer. As shown in Table 1, the AR baseline achieves superior accuracy and lower cumulative uncertainty, highlighting that greedy MDM samplers fail by prematurely committing to low-uncertainty answer tokens before the reasoning is established.

*Table 1.* Quantitative results for Case Study 2.

| Metric | Entropy | Confidence | Margin | AR |
|---|---|---|---|---|
| $\tilde{H} \downarrow$ | 32.75 | 31.68 | 35.60 | **25.19** |
| Acc. (%) $\uparrow$ | 67 | 73 | 66 | **90** |

> **Observation 1:** Existing greedy certainty-based samplers often fail to find near-optimal decoding paths.

As illustrated in Figure 1, an optimal decoding action should be evaluated not only by its own prediction certainty but also by the *information gain* it provides for the remainder of the generation process. To address the limitations of existing samplers, it is essential to account for this information gain when making current decisions. As shown in Figure 1, our proposed Info-Gain Sampler effectively addresses this one-way challenge by prioritizing the decoding of factors with an 84% probability.

In standard ARMs, assessing information gain typically requires computationally expensive techniques, such as Monte Carlo Tree Search, to simulate future trajectories. This is primarily due to the next-token bottleneck: since ARMs only provide the probability of the immediate next token, multi-step look-ahead becomes prohibitively slow.

> **Question 2:** Can Masked Diffusion Models efficiently assess information gain of a decoding action?

Unlike ARMs, MDMs do not have the next-token bottleneck. MDMs leverage bidirectional attention, allowing the model to simultaneously evaluate the impact of any decoding action on the uncertainty of the entire sequence. This architectural advantage enables the model to "see" how fill-

ing a mask affects the uncertainty of all remaining masks in a single forward pass.

> **Observation 2:** MDMs' bidirectional architecture enables efficient information gain estimation in one forward pass, bypassing expensive iterative computations.

## 3.2. Information Gain Sampler

The insights from **Observation 1** and **Observation 2** suggest that greedy optimization alone is insufficient for minimizing cumulative uncertainty across the entire sequence.

We introduce the **Information Gain (Info-Gain) Sampler** which leverages the bidirectional nature of MDMs to balance the immediate uncertainty cost of a decoding decision against its expected information gain over the remaining masked positions.

### 3.2.1. OBJECTIVE OF INFO-GAIN SAMPLER

We first define **state uncertainty** as the average marginal entropy over the masked positions in state $z_t$:

$$\mathcal{H}(z_t) = \frac{1}{|\mathcal{M}_t|} \sum_{\ell \in \mathcal{M}_t} H^{(\ell)}(z_t) \qquad (3)$$

The state uncertainty quantifies the information remaining to be resolved by the model and can be computed efficiently via a single forward pass.

The **information gain** of action $a_t$ is defined as the reduction in state uncertainty (equivalently, the decrease in marginal entropy over the remaining masked positions) it induces:

$$\mathrm{IG}(a_t; z_t) := \mathcal{H}(z_t) - \mathcal{H}(z_{t-1}) \qquad (4)$$

where $z_{t-1} = \mathrm{Apply}(z_t, a_t)$ denotes the state obtained after executing action $a_t$ from state $z_t$.

The total impact of a decoding action $a_t$ is thus decomposed into two components:

**(1) Immediate Cost:** the uncertainty of the tokens being decoded in the current step, measured by the sum of marginal entropy over the chosen positions $C(a_t \mid z_t)$.

**(2) Information Gain:** the reduction in the uncertainty over the remaining mask positions, quantified by the information gain $\mathrm{IG}(a_t; z_t)$.

To balance these two components, we define the Info-Gain Sampler objective as:

$$J_{\mathrm{IG}}(a_t \mid z_t) = \underbrace{\mathrm{IG}(a_t; z_t)}_{\text{Information Gain}} - \underbrace{C(a_t \mid z_t)}_{\text{Immediate Cost}}, \qquad (5)$$

We provide further theoretical analysis in Appendix C.

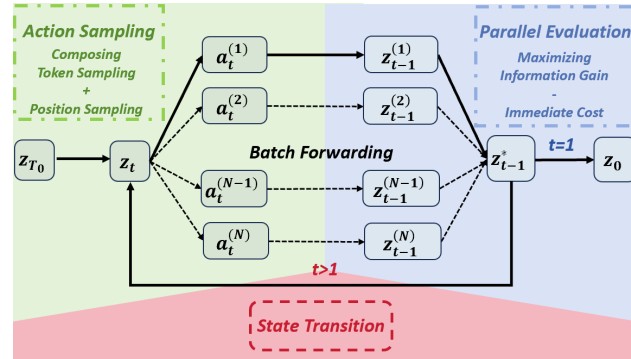

*Figure 2.* The Info-Gain Sampler workflow. Starting from state $z_{T_0}$, the sampler iteratively: (1) samples candidate actions, (2) evaluates $J_{\mathrm{IG}} = $ Information Gain $-$ Immediate Cost in parallel to select the optimal successor state $z_{t-1}^*$, and (3) executes the state transition until reaching the final sequence $z_0$.

### 3.2.2. IMPLEMENTATION OF INFO-GAIN SAMPLER

**Action Sampler.** Following previous work (Peng et al., 2025; Yang et al., 2025), we explore the large action space by generating a candidate set $\mathcal{C}$ of size $N$ through a two-stage sampling process: (1) *Token Sampling*: drawing tokens $v_\ell$ from $p_\theta$ with token temperature $\tau_{\text{token}}$, and (2) *Position Sampling*: selecting positions $\ell \in \mathcal{M}_t$ using a softmax over certainty scores $\phi(\ell, z_t)$ with position temperature $\tau_{\text{pos}}$. Each candidate action $a_t = \{(\ell, v_\ell)\}$ is formed by pairing these samples, providing a diverse and high-quality set for evaluation. The size of $a_t$ is determined by a step scheduling function.

At each decoding step, Info-Gain Sampler follows a three-step cycle to determine and execute the most informative action (Figure 2):

**(1) Sampling:** We sample a candidate set $\mathcal{C} = \{a_t^{(1)}, \ldots, a_t^{(N)}\}$ of diverse actions using the *Action Sampler*. This step explores the combinatorially large action space by proposing multiple potential actions.

**(2) Evaluation:** We compute the objective $J_{\mathrm{IG}}(a_t \mid z_t)$ for all candidates $a_t$ in the set $\mathcal{C}$. Crucially, as noted in the previous section, this evaluation is highly efficient as it requires only a single batched forward pass to estimate the future information gain for all candidates simultaneously.

**(3) Transition:** The optimal action is selected as $a_t^* = \arg\max_{a \in \mathcal{C}} J_{\mathrm{IG}}(a \mid z_t)$. We then execute this action to transition to the next state $z_{t-1}^*$, repeating the cycle until all masked positions are filled and a complete sequence is generated.

### 3.2.3. EFFICIENT IMPLEMENTATION OF INFO-GAIN SAMPLER

To ensure efficiency, candidate evaluations are performed in parallel within a single batched forward pass. We further optimize the sampler by restricting the information-gain computation to the current active block $\mathcal{B}$ (Arriola et al., 2025). Following the block-wise approximate KV-cache strategy of Fast-dLLM (Wu et al., 2025b), tokens outside the active block are unchanged during candidate evaluation and can therefore be reused through a shared dual cache: the prefix cache stores tokens before $\mathcal{B}$, while the suffix cache stores the unchanged masked context after $\mathcal{B}$. Additionally, we implement a high-confidence bypass: if the maximum token probability exceeds a threshold $\gamma$, the corresponding positions are directly fixed into the action set. If the number of such high-confidence positions exceeds the predefined size of the action set for the current step, only the top-$k$ positions with the highest confidence are selected for decoding. This hybrid approach, inspired by Wu et al. (2025b), significantly reduces inference latency while preserving planning quality. Because Info-Gain effectively reduces uncertainty during decoding, the high-confidence bypass is triggered more frequently, making the mechanism exceptionally efficient.

### 3.2.4. WHY IS INFO-GAIN EFFECTIVE

**State uncertainty** can be viewed as a proxy for whether the current decoding state lies close to the **training data manifold**, where logically coherent and fluent text is more likely to reside: low uncertainty reflects a concentrated predictive distribution, while high uncertainty signals potential deviation into poorly conditioned regions associated with inconsistent or unnatural expressions. Unlike greedy certainty-based samplers that cannot recognize such deviation signals because locally certain actions may still increase future uncertainty, the **Info-Gain Sampler** explicitly uses state uncertainty through its information-gain term: actions that increase uncertainty negatively affect the information-gain objective, thereby discouraging their selection. This mechanism helps the Info-Gain Sampler maintain logical coherence and fluency throughout decoding, even under high sampling temperatures.

## 4. Experiments

### 4.1. Experimental Setup

**Benchmarks.** We evaluate the effectiveness of the Info-Gain Sampler on diverse settings: (1) Fully-Attention MDM Reasoning: Math (GSM8K, MATH-500; 0-shot) (Cobbe et al., 2021; Hendrycks et al., 2021), Code (HumanEval, MBPP; 0-shot) (Chen et al., 2021; Austin et al., 2021b), and Planning (4×4 Sudoku; 5-shot (Qin et al., 2025),

Countdown; 3-shot with 3 numbers (Ye et al., 2025)), reporting average Pass@1 accuracy over 5 runs; (2) Semi-Autoregressive MDM Reasoning: evaluated on the same Math and Code benchmarks in a zero-shot setting, reporting average Pass@1 accuracy over 5 runs; (3) Multimodal Text-to-Image Generation: evaluated on ImageNet-512 and GenEval (Ghosh et al., 2023), with IS (Salimans et al., 2016), FID (Heusel et al., 2017), and sFID (Radford et al., 2018; Salimans et al., 2016) reported for ImageNet-512, and attribute-wise scores for GenEval; (4) Creative Writing: evaluated on AlpacaEval (Li et al., 2023), using an LLM-as-a-judge to compute the length-controlled win rate (Dubois et al., 2024) against baselines following past works (Nguyen et al., 2024).

**Models.** For reasoning tasks, we use Dream-7B-Instruct (Ye et al., 2025) as the fully-attention representative, and SDAR-8B-Chat (Cheng et al., 2025) and TraDo-8B-Instruct (Wang et al., 2025) for block diffusion settings using KV-cache following (Wu et al., 2025b). For image generation, we employ the MMaDa (Yang et al., 2025) model. For creative writing, we employ the SDAR-8B-Chat model.

**Baselines.** We compare Info-Gain Sampler against several sampling baselines, including *Uniform* (Nie et al., 2025), *Autoregressive (AR)*, *Entropy* (Ye et al., 2025), *Confidence* (Chang et al., 2022), *Margin* (Kim et al., 2025a), *KLASS* (Kim et al., 2025b), and *PC-Sampler* (Huang et al., 2025). We include a concurrent work, *LookUM* (Lee et al., 2025), which also employs a look-ahead mechanism. We adapt their method to keep it as close as possible to the Info-Gain Sampler, enabling a fairer comparison. Detailed descriptions of these baselines are provided in Appendix A. For math and code tasks, we adopt a block diffusion approach, as previous studies (Arriola et al., 2025; Wu et al., 2025b) have shown that this can significantly improve performance. For planning-centric tasks, we remove positional decoding constraints by setting the block size equal to the total generation length, allowing for global optimization across the entire sequence.

**Hyperparameters.** For reasoning tasks, we employ a position temperature $\tau_{\text{pos}} = 0.1$ and $N = 8$ candidates for the Info-Gain Sampler, with the acceleration threshold $\gamma$ set to $0.8$. We evaluate the performance under both $K = 1$ and $K = 2$ tokens per step settings. For text-to-image experiments, we set $\tau_{\text{pos}} = 0.4$ and $N = 8$ with a 50-step cosine scheduler. Detailed settings for all benchmarks and baseline-specific parameters are provided in Appendix B.

### 4.2. Results and Analysis

**Reasoning on Full-Attention MDMs.** As shown in Table 2, the Info-Gain Sampler consistently outperforms all baselines on Dream-7B-Instruct while using the acceleration techniques introduced in Section 3.2.3 to keep the additional

*Table 2.* Performance on the full-attention MDM (**Dream-7B-Instruct**). Results are reported with decoding rates $K \in \{1, 2\}$, where reasoning tasks(GSM8K, MATH500, HumanEval, MBPP) use a block size of 16 and planning tasks (Sudoku, Countdown) are decoded globally. We report the accuracy for each task, along with the average accuracy (Avg.) and cumulative entropy ($\tilde{H}$) per generation.

| K | Sampler | GSM8K | MATH500 | HumanEval | MBPP | Sudoku | Countdown | Avg. | $\tilde{H} \downarrow$ |
|---|---|---|---|---|---|---|---|---|---|
| | Uniform | 18.7 | 14.3 | 14.3 | 18.6 | 52.8 | 7.6 | 21.1 | 389.2 |
| | AR | 53.8 | 23.0 | 23.7 | 22.2 | 38.2 | 25.8 | 31.1 | 273.1 |
| | Entropy | 55.8 | 27.0 | 26.2 | 23.2 | 76.0 | 23.2 | 38.6 | 247.8 |
| | Confidence | 61.9 | 29.4 | 26.8 | 25.2 | 81.6 | 36.2 | 43.5 | 249.2 |
| 2 | Margin | 65.5 | 28.8 | 28.8 | 23.6 | 74.4 | 28.9 | 41.7 | 287.5 |
| | KLASS | 67.3 | 30.4 | 31.9 | 30.1 | 82.1 | 35.4 | 46.2 | 239.3 |
| | PC-Sampler | 72.8 | 32.3 | 36.4 | 34.4 | 80.2 | 42.4 | 49.8 | 158.3 |
| | LookUM | 75.2 | **35.2** | 38.4 | 36.1 | 77.4 | 39.2 | 50.3 | 134.9 |
| | **Info-Gain** | **77.7** | 34.2 | **42.9** | **39.4** | **82.2** | **44.1** | **53.4** | **104.3** |
| | Uniform | 30.4 | 15.8 | 16.7 | 30.8 | 60.4 | 8.8 | 27.2 | 199.3 |
| | AR | 76.5 | 43.8 | 41.9 | 35.6 | 61.8 | 35.5 | 49.2 | 121.7 |
| | Entropy | 75.7 | 47.0 | 46.4 | 45.0 | 78.8 | 33.6 | 54.4 | 95.6 |
| | Confidence | 78.8 | 46.6 | 49.4 | 39.8 | 81.5 | 39.2 | 55.9 | 99.5 |
| 1 | Margin | 77.5 | 46.5 | 41.5 | 40.4 | 80.3 | 35.2 | 53.6 | 107.3 |
| | KLASS | 78.2 | 47.2 | 34.2 | 40.6 | 79.9 | 34.3 | 52.4 | 99.6 |
| | PC-Sampler | 81.5 | 46.8 | 54.4 | 46.2 | 83.6 | 41.8 | 59.1 | 78.4 |
| | LookUM | 78.3 | **51.4** | 52.3 | 45.9 | 78.4 | 43.3 | 58.2 | 61.4 |
| | **Info-Gain** | **83.3** | 51.3 | **59.2** | **48.4** | **84.4** | **45.2** | **62.0** | **48.6** |

*Table 3.* Performance on semi-autoregressive MDMs (**SDAR-8B-Chat** and **TraDo-8B-Instruct**). Results are reported with a block size of 16 and token temperature $\tau_{\text{token}} = 0.7$. We report the accuracy for each task, along with the average accuracy (Avg.) and cumulative entropy ($\tilde{H}$) per generation. The Info-Gain Sampler consistently achieves the strongest performance across all settings.

| | | SDAR-8B-Chat | | | | | | TraDo-8B-Instruct | | | | | |
|---|---|---|---|---|---|---|---|---|---|---|---|---|---|
| K | Sampler | GSM8K | MATH500 | HumanEval | MBPP | Avg. | $\tilde{H} \downarrow$ | GSM8K | MATH500 | HumanEval | MBPP | Avg. | $\tilde{H} \downarrow$ |
| | Entropy | 42.2 | 24.4 | 26.2 | 20.6 | 28.4 | 238.6 | 31.9 | 17.0 | 20.7 | 21.8 | 22.8 | 419.5 |
| | Confidence | 47.2 | 36.6 | 24.4 | 20.2 | 32.1 | 204.1 | 36.5 | 39.2 | 19.5 | 22.0 | 29.3 | 334.1 |
| 2 | Margin | 45.2 | 22.4 | 19.5 | 19.8 | 26.7 | 230.9 | 33.2 | 17.0 | 18.9 | 21.8 | 22.7 | 398.0 |
| | KLASS | 50.4 | 32.3 | 30.7 | 26.6 | 35.0 | 210.3 | 50.4 | 32.3 | 30.7 | 26.6 | 35.0 | 334.3 |
| | LookUM | 75.3 | 44.9 | 28.2 | 31.8 | 45.0 | 103.2 | 79.8 | 40.4 | 46.8 | 33.9 | 50.2 | 119.2 |
| | **Info-Gain** | **82.7** | **54.6** | **46.3** | **39.4** | **55.8** | **74.1** | **83.9** | **40.9** | **58.8** | **37.4** | **55.3** | **98.0** |
| | Entropy | 68.8 | 44.6 | 37.8 | 49.0 | 50.0 | 120.4 | 63.9 | 36.4 | 37.8 | 42.4 | 45.2 | 171.4 |
| | Confidence | 67.9 | 51.4 | 42.1 | 46.2 | 51.9 | 117.4 | 64.5 | 55.4 | 40.2 | 47.6 | 51.9 | 163.5 |
| 1 | Margin | 65.3 | 40.2 | 32.3 | 43.2 | 45.3 | 138.2 | 62.3 | 36.4 | 37.2 | 42.4 | 44.5 | 208.0 |
| | KLASS | 69.9 | 42.3 | 45.7 | 46.6 | 51.1 | 105.3 | 65.4 | 40.8 | 40.9 | 47.0 | 48.5 | 180.3 |
| | LookUM | 80.3 | 60.0 | 38.2 | 39.8 | 54.6 | 53.7 | 88.0 | 46.2 | 43.4 | 49.8 | 56.9 | 60.8 |
| | **Info-Gain** | **87.9** | **61.8** | **62.2** | **53.0** | **66.2** | **41.0** | **88.4** | **62.8** | **67.4** | **54.0** | **68.2** | **52.1** |

*Table 4.* Text-to-Image results on **GenEval** and **ImageNet-512** with token temperature $\tau_{\text{token}} = 0.4$. The Info-Gain Sampler demonstrates superior alignment and fidelity, significantly outperforming all baselines across multimodal generation metrics.

| Method | GenEval | | | | | | | ImageNet-512 | | |
|---|---|---|---|---|---|---|---|---|---|---|
| | single obj.↑ | two obj. ↑ | count ↑ | colors ↑ | pos ↑ | attr ↑ | Avg ↑ | IS ↑ | FID ↓ | sFID ↓ |
| Uniform | 94.1 | 66.7 | 38.4 | 78.2 | 19.0 | 28.8 | 54.2 | 49.3 | 46.8 | 123.9 |
| Entropy | 94.3 | 67.3 | 46.0 | 79.9 | 17.8 | 26.8 | 55.3 | 52.4 | 44.8 | 93.4 |
| Confidence | 93.8 | **69.7** | 46.3 | **81.9** | 16.0 | 27.0 | 56.0 | 53.3 | 43.3 | 92.0 |
| Margin | 94.0 | 68.7 | 47.3 | 80.1 | 19.0 | 29.0 | 56.3 | 51.9 | 45.2 | 95.3 |
| **Info-Gain** | **97.5** | 68.7 | **47.5** | 79.8 | **25.0** | **32.0** | **58.2** | **63.0** | **38.1** | **83.7** |

overhead modest. In particular, Info-Gain Sampler delivers substantial gains in average accuracy, surpassing the best-performing baselines by 3.1 percentage points at $K = 2$ and 2.9 percentage points at $K = 1$. Experimental results

*Table 5.* Win rate of Info-Gain Sampler against baselines (%)

| Temperature | K | Win rate vs. baseline (%) | | | |
|---|---|---|---|---|---|
| | | AR | Confidence | Entropy | Margin |
| 0.5 | 1 | 60.1 | 65.8 | 59.1 | 63.6 |
| | 2 | 63.9 | 68.9 | 70.4 | 64.7 |
| 1.0 | 1 | 54.8 | 57.7 | 60.1 | 55.2 |
| | 2 | 65.2 | 61.1 | 65.7 | 57.5 |
| 1.5 | 1 | 58.4 | 53.0 | 60.3 | 54.6 |
| | 2 | **69.7** | **70.1** | **80.3** | **66.8** |

further demonstrate that Info-Gain Sampler attains a significantly lower cumulative entropy $\tilde{H}$, reaching only 65.9% ($K = 2$) and 62.0% ($K = 1$) of the best-performing greedy selection baseline (PC-Sampler (Huang et al., 2025))—underscoring its ability to discover more globally optimized trajectories. Compared to LookUM (Lee et al., 2025), a concurrent baseline that incorporates a look-ahead term, our method achieves superior performance on Code and Planning tasks. These tasks demand high token-level precision, where our immediate cost term proves more effective in mitigating local errors.

**Reasoning on Semi-AR MDMs.** Results for Semi-AR models (Table 3) further validate the robustness of Info-Gain Sampler. Notably, while introducing a non-zero token temperature ($\tau_{\text{token}} = 0.7$) degrades the performance of baselines, Info-Gain Sampler maintains a substantial lead, outperforming the best baseline by 11.6 and 11.3 percentage points in average accuracy under $K = 1$ settings for SDAR-8B-Chat and TraDo-8B-Instruct, respectively. The consistent reduction in $\tilde{H}$ across different architectures underscores the universal effectiveness of our information-gain-based objective.

**Text-to-Image Generation.** In multimodal settings (Table 4), Info-Gain Sampler excels in both alignment and fidelity. It achieves the highest average GenEval score (58.2 vs. 56.3 for Margin) and significantly improves "positional" (25.0 vs. 19.0) and "attribute" (32.0 vs. 29.0) sub-scores. Furthermore, on ImageNet-512, Info-Gain Sampler substantially improves FID (from 43.3 to 38.1) and IS (from 53.3 to 63.0), demonstrating its broad generalizability on multimodal generation tasks.

**Creative Writing.** For creative writing (Table 5), the Info-Gain Sampler consistently outperforms all baselines across various token temperatures $\tau_{\text{token}}$. At a high temperature of $\tau_{\text{token}} = 1.5$, where increased stochasticity often degrades coherence, our sampler achieves a peak win rate of **80.3%** against the Entropy baseline. By prioritizing informative actions through its look-ahead mechanism, the Info-Gain Sampler exhibits superior robustness to temperature scaling in MDMs, effectively balancing creativity and coherence. Across all settings and baselines, it maintains an average win

rate of **62.8%**, demonstrating that introducing information gain enhances both creative diversity and textual coherence under temperature variations.

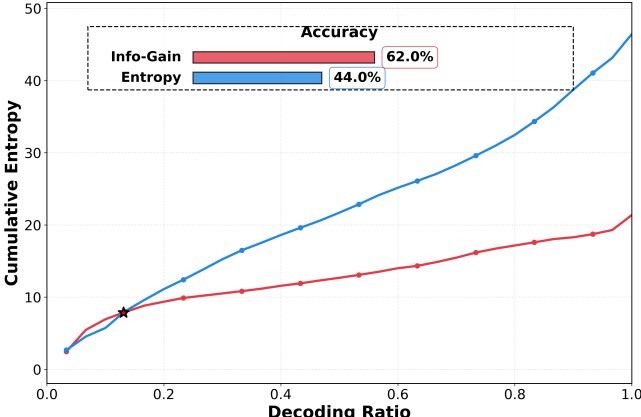

*(a)* Cumulative entropy trajectories

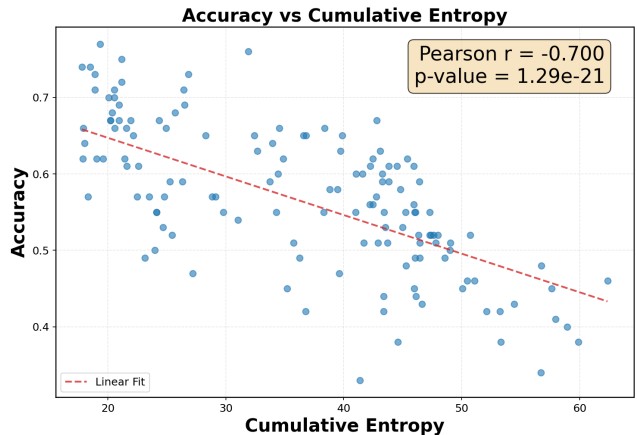

*(b)* Accuracy vs. Cumulative Entropy

*Figure 3.* **Analysis of Cumulative Entropy.** (a) Cumulative entropy trajectories for the Entropy baseline and Info-Gain Sampler on a synthetic set of 100 simple arithmetic problems that can be answered within a short window. We use global decoding with a fixed length of 64 tokens. (b) Correlation between average accuracy and average cumulative entropy across various sampling configurations.

### 4.3. Ablation Study

**Optimization of Cumulative Uncertainty.** As shown in Table 2, the **Info-Gain Sampler** significantly outperforms baselines in optimizing cumulative uncertainty. By tracking cumulative entropy during decoding on mathematical reasoning tasks (Fig. 3), we observe that: (1) The Info-Gain heuristic balances immediate cost with future gains, yielding non-linear entropy growth that stabilizes earlier than the greedy Entropy baseline. (2) Cumulative entropy shows a strong **negative correlation with accuracy** (Pearson's $r = -0.70$), validating it as a reliable proxy for decoding quality.

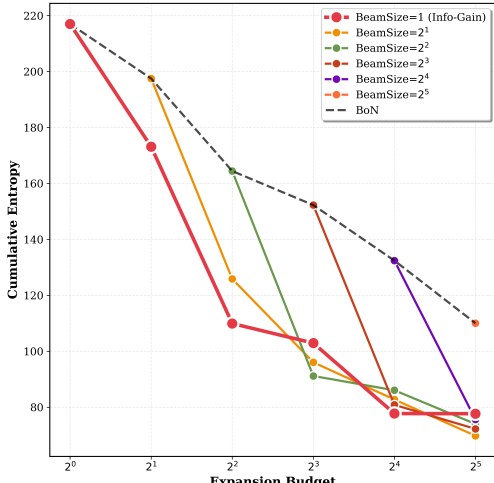

*Figure 4.* Impact of different beam sizes on the MATH-500 dataset. Specifically: **Beam Size = 1** is a special case equivalent to the Info-Gain Sampler; **Beam Size = Expansion Budget** is equivalent to the Best-of-$N$ (BoN) baseline; and **Intermediate Values** represent a look-ahead beam search algorithm using Info-Gain as the pruning heuristic.

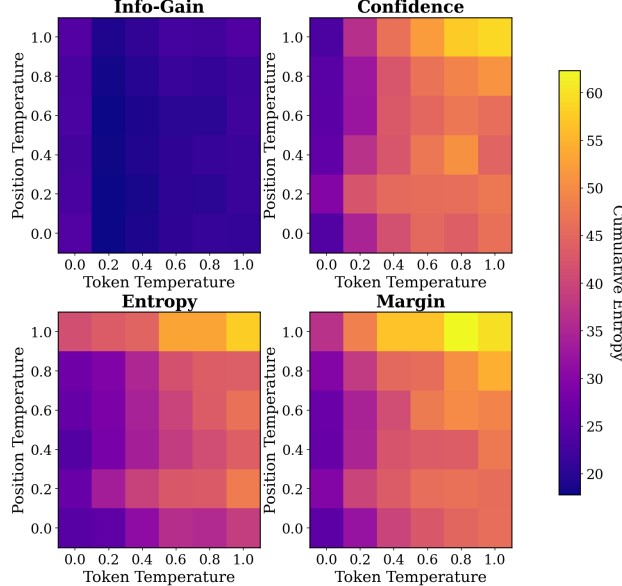

*Figure 5.* **Temperature Sensitivity.** Cumulative trajectory uncertainty under varying position and token temperatures on the 100 simple arithmetic problems, evaluated using global decoding with a fixed length of 64 tokens.

**Comparison of Info-Gain Variants.** We compare the **Info-Gain Sampler** ($B = 1$), **Info-Gain Beam Search** ($B > 1$), and **Best-of-N (BoN)** under a fixed computational budget (Figure 4). For beam search variants, we set the beam width to $B$ and rank partial hypotheses using the accumulated Info-Gain objective plus the current state-uncertainty score. The decoding process terminates when all hypotheses in the beam have been fully decoded. (1) The **Info-Gain Sampler**

($B = 1$) performs near the Pareto frontier, achieving near-optimal results while remaining highly parallelizable and avoiding complex KV-cache management. (2) Both Info-Gain variants significantly outperform **BoN**, proving that global planning via information gain is superior to simply increasing independent samples. (3) Increasing **Beam Size** under given expansion budget yields marginal uncertainty reduction but incurs higher memory overhead (Appendix F.1).

**Compatibility with Temperature Sampling.** We investigate the impact of position and token temperature settings on cumulative uncertainty. For the baselines, the position temperature mechanism is implemented by applying a softmax with temperature to the heuristic scores followed by categorical sampling, where a position temperature of zero corresponds to the original greedy sampling. As shown in Figure 5, the Info-Gain Sampler maintains stable, low trajectory uncertainty across various temperature scales without sensitive tuning. Importantly, low cumulative entropy reflects more optimized decoding rather than mode collapse, as evidenced by the preserved diversity and competitive win rates in creative writing (Table 5). In contrast, other baselines are highly sensitive to temperature changes, leading to decoding instability.

## 5. Limitation and Future Work

While the Info-Gain Sampler demonstrates significant improvements in generation quality across multiple domains, there are several avenues for further refinement.

**More Efficient Implementation.** Although the Info-Gain Sampler leverages parallel evaluation and acceleration techniques to minimize latency, the search process still incurs higher computational cost than greedy decoding. Future work could explore more efficient lookahead mechanisms, adaptive branching, and hardware-level optimizations to further enhance inference throughput.

**Refinement of Action Sampler.** Our action sampler currently leverages local uncertainty as a heuristic for candidate generation. Although this approach is robust and yields high-quality plans, future research could investigate more sophisticated sampling strategies that go beyond local heuristics. Such advancements would likely enhance both the diversity and quality of the candidate set.

## 6. Related Work

**Masked Diffusion Models.** Discrete diffusion models provide a non-autoregressive alternative for sequence generation, grounded in theoretical frameworks for discrete and masked data (Austin et al., 2021a; Lou et al., 2024; Rombach et al., 2022). Unlike ARMs, which often require training interventions for long-horizon planning (Hu et al., 2025; Teoh et al., 2025), bidirectionality in MDMs naturally en-

*Table 6.* Unified view of MDM samplers. **Selection Criteria** denotes the objective function optimized at each decoding step. **Temperature Sensitivity** indicates robustness of performance to temperature-based stochastic sampling, while **Greedy Location Selection** marks whether the sampler selects positions based solely on immediate certainty without considering future information gain.

| Sampler | Selection Criteria | Temperature Sensitivity | Greedy Selection |
|---|---|---|---|
| Uniform (Austin et al., 2021a) | $-$ | High | ✗ |
| Confidence (Chang et al., 2022) | $\sum_{\ell \in A_t} \max p_\theta(x_\ell \mid z_t)$ | High | ✓ |
| Entropy (BenHamu et al., 2025) | $-\sum_{\ell \in A_t} H^{(\ell)}(z_t)$ | High | ✓ |
| Margin (Kim et al., 2025a) | $\sum_{\ell \in A_t} (p_{\text{top1}} - p_{\text{top2}})$ | High | ✓ |
| KLASS[†] (Kim et al., 2025b) | $\sum_{\ell \in A_t} (\max p_\theta(x_\ell \mid z_t) + \mathbf{1}_{D_{KL} < \epsilon})$ | High | ✓ |
| PC-Sampler (Huang et al., 2025) | $\sum_{\ell \in A_t} (w_\ell \cdot \mathcal{C}_t^{(\ell)})$ | Moderate | ✓ |
| LookUM (Lee et al., 2025) | $\frac{1}{|\mathcal{M}_{t-1}|} \cdot \sum_{\ell \in \mathcal{M}_{t-1}} \phi(z_{t-1}, \ell)$ | Moderate | ✗ |
| **Info-Gain (Ours)** | $\text{IG}(a_t; z_t) - \sum_{\ell \in A_t} H^{(\ell)}(z_t)$ | **Low** | ✗ |

[†]We adapt KLASS to ensure its decoding procedure adheres to the specified step scheduler.

ables global reasoning (Ye et al., 2024). MDM research has diverged into training large models from scratch, like LLaDA (Nie et al., 2025), and adapting pre-trained autoregressive models, such as Dream, DiffuLLaMA, and DIMPLE (Ye et al., 2025; Gong et al., 2025; Yu et al., 2025). To enhance long-sequence modeling, hybrid semi-autoregressive (Semi-AR) architectures like Block Diffusion (Arriola et al., 2025), Diffusion Forcing (Chen et al., 2024), and absorbing diffusion variants (Zheng et al., 2025) enable KV-caching for better efficiency. Recent optimizations like Fast-dLLM/v2 (Wu et al., 2025b;a) and models such as SDAR (Cheng et al., 2025), TraDo (Wang et al., 2025), and WeDLM (Liu et al., 2025) have further advanced complex reasoning and long-text generation.

**Samplers.** Due to the causal factorization, sampling strategies for ARMs typically rely on assessing and regulating local uncertainty in next-token prediction to improve generation quality and diversity. A broad class of samplers has been proposed, including deterministic decoding strategies such as beam search (Wu et al., 2016; Freitag & Al-Onaizan, 2017) and stochastic decoding strategies (Holtzman et al., 2019; Fan et al., 2018; Nguyen et al., 2024). In contrast to ARMs, MDMs introduce a fundamentally different sampling problem: beyond deciding *what* token to decode, samplers must also decide *where* to decode within the non-causal sequence. This expanded decision space amplifies the impact of early decoding choices and renders local uncertainty criteria insufficient. Still, existing MDM samplers typically rely on greedy, certainty-based heuristics to select decoding positions, using metrics such as entropy and margin scores (Nie et al., 2025; Ye et al., 2025; Chang et al., 2022). Some approaches further incorporate calibration or stability refinements to improve robustness (BenHamu et al., 2025; Kim et al., 2025b; Huang et al., 2025). Nevertheless, these approaches share a common limitation: decoding decisions are made myopically based on greedy metrics. They do not account for the downstream impact of each decoding decision on global uncertainty or information gain across the remaining masked tokens. We provide a comparison between existing samplers for MDMs in Table 6.

## 7. Conclusion

We propose the **Info-Gain Sampler**, a training-free decoding framework for Masked Diffusion Models that exploits bidirectional attention to incorporate information gain in action selection, balancing immediate certainty with future uncertainty reduction and mitigating the myopia of uncertainty-based samplers. Across reasoning, code generation, planning, and image generation tasks, Info-Gain Sampler consistently improves performance, achieving gains of 2.9–11.6 percentage points in average reasoning accuracy across full-attention and semi-autoregressive settings, a 62.8% average win rate in creative writing, and an improvement of 1.9 percentage points on GenEval. It remains compatible with both full-attention and semi-autoregressive architectures, reduces **cumulative uncertainty**, and offers a principled bridge from local heuristics to global planning for non-autoregressive generation.

## Impact Statement

This paper presents work whose goal is to advance the field of Machine Learning. There are many potential societal consequences of our work, none which we feel must be specifically highlighted here.

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

# A. Formal Definitions of Baseline Samplers

To provide a rigorous comparison, we formalize the action selection mechanism for each baseline sampler. At each decoding step $t$, let $p_\theta(\cdot \mid z_t, \ell)$ denote the predicted token distribution at masked position $\ell \in \mathcal{M}_t$. The samplers differ in their scoring function $\phi(z_t, \ell)$, where the top-$K$ positions with the highest scores are selected for decoding.

**Uniform (Nie et al., 2025)**   This baseline selects positions uniformly at random from the mask set $\mathcal{M}_t$:

$$\phi_{\text{Uniform}}(z_t, \ell) = \epsilon, \quad \epsilon \sim \mathcal{U}(0, 1) \tag{6}$$

**Confidence (Chang et al., 2022)**   This baseline prioritizes positions where the model is most certain about the top-1 prediction:

$$\phi_{\text{Conf}}(z_t, \ell) = \max_{v \in \mathcal{V}} p_\theta(v \mid z_t, \ell) \tag{7}$$

**Entropy (Ye et al., 2025)**   Positions with the minimum predictive uncertainty are selected:

$$\phi_{\text{Entropy}}(z_t, \ell) = -H(X^{(\ell)} \mid z_t) = \sum_{v \in \mathcal{V}} p_\theta(v \mid z_t, \ell) \log p_\theta(v \mid z_t, \ell) \tag{8}$$

**Margin (Kim et al., 2025a)**   This baseline considers the gap between the two most likely tokens, selecting positions with the largest margin:

$$\phi_{\text{Margin}}(z_t, \ell) = p_{\text{top1}} - p_{\text{top2}} \tag{9}$$

**KLASS (Kim et al., 2025b)**   This method incorporates a KL-divergence constraint to maintain consistency between consecutive decoding steps. Originally, KLASS (Kim et al., 2025b) supports dynamic decoding by selecting all positions that satisfy the KL threshold. To ensure a fair performance comparison with other samplers, we adapt it to decode a fixed number of tokens per step. Following the original implementation, we set the threshold $\epsilon = 5 \times 10^{-4}$. Effectively, KLASS prioritizes positions where the KL divergence is below the threshold, ranking them by confidence, and only falls back to other positions if more tokens are required. This can be formalized as the following scoring function:

$$\phi_{\text{KLASS}}(z_t, \ell) = \phi_{\text{Conf}}(z_t, \ell) + \mathbb{1}[D_{\text{KL}}(p_\theta(\cdot \mid z_t, \ell) \| p_\theta(\cdot \mid z_{t-1}, \ell)) < \epsilon] \tag{10}$$

where positions with higher scores are prioritized. In our experiments, we first select positions satisfying the KL constraint based on confidence, and then fill the remaining budget from the other positions.

**PC-Sampler (Huang et al., 2025)**   This sampler regulates the decoding trajectory by combining a position-aware weight with content-aware confidence calibration. It modulates the selection priority of each candidate token $x_\ell$ at position $\ell$ using an exponential decay function $w_\ell = e^{-\lambda \cdot \ell}$, where $\lambda \geq 0$ controls the positional penalty. To discourage generic tokens, it calibrates the confidence score using the token frequency distribution $p_{\mathcal{D}'}(x_\ell)$ and a content-aware calibration term $\mathcal{C}_t^{(\ell)}$:

$$\phi_{\text{PC}}(z_t, \ell) = \mathcal{C}_t^{(\ell)} \cdot w_\ell \tag{11}$$

**Block Diffusion (Arriola et al., 2025; Wu et al., 2025b)**   This sampler employs a position scheduling function $I_t(\ell)$ that indicates whether position $\ell$ lies within the currently active diffusion block at time step $t$. Specifically, $I_t(\ell)$ takes the value 1 if $\ell \in \mathcal{B}_{\sigma(t)}$ and 0 otherwise, where $\mathcal{B}_{\sigma(t)}$ denotes the active block determined by a scheduling rule $\sigma(t)$. The sampling process restricts candidate positions to the active block and applies a standard heuristic:

$$\phi_{\text{Block}}(z_t, \ell) = \begin{cases} \phi(z_t, \ell), & \text{if } I_t(\ell) = 1, \\ -\infty, & \text{otherwise.} \end{cases} \tag{12}$$

Here, the active block $\mathcal{B}_{\sigma(t)}$ slides over time according to $\sigma(t)$, typically following a sequential or random traversal of the position indices, thereby gradually diffusing information across the entire sequence.

**LookUM (Lee et al., 2025)** This framework improves decoding in masked diffusion language models by selecting optimal token unmasking orders during inference. Unlike myopic heuristics that consider only immediate next steps, LookUM generates multiple candidate unmasking trajectories (paths) and selects the most promising one based on global sequence-level certainty. This can be formalized as the following scoring function:

$$\phi_{\text{LookUM}}(z_t, \ell) = \frac{1}{|\mathcal{M}_{t-1}^{(\ell)}|} \sum_{j \in \mathcal{M}_{t-1}^{(\ell)}} \phi(z_{t-1}^{(\ell)}, j) \tag{13}$$

where $z_{t-1}^{(\ell)}$ denotes the state obtained after tentatively unmasking position $\ell$, $\mathcal{M}_{t-1}^{(\ell)}$ denotes its remaining masked positions, and $\phi(\cdot)$ is a base metric that can be instantiated as negative entropy, confidence, or margin. Following the empirical findings in the paper, we adopt negative entropy as our specific implementation of $\phi(\cdot)$ due to its demonstrated effectiveness in capturing sequence-level uncertainty. We omit the design of SMC and NIS in LookUM (Lee et al., 2025) to enable a clear comparison, as they require task-specific parameter tuning.

## B. Detailed Hyperparameter Settings

In this section, we provide a detailed overview of the hyperparameter settings used across all experiments. All experiments were conducted on NVIDIA A800 GPUs.

**Reasoning and Coding Tasks.** For tasks involving logical reasoning (GSM8K, MATH500) and code generation (HumanEval, MBPP), we set the token sampling temperature $\tau_{\text{token}} = 0.7$ to strike a balance between output diversity and structural coherence. For the Info-Gain Sampler, we employ a relatively low position temperature $\tau_{\text{pos}} = 0.1$ to focus the selection on the most informative candidate positions, while generating $N = 8$ candidate actions per step for parallel evaluation. To maintain high throughput during predictable decoding phases, we set the acceleration threshold $\gamma = 0.8$, which allows the sampler to skip the full evaluation routine when the maximum token probability is high. For block diffusion settings on Semi-AR models (SDAR and TraDo), we use a fixed block size of 16. The decoding budget $K$ (tokens per step) is varied between 1 and 2 to evaluate performance under different acceleration ratios. Maximum generation lengths are benchmark-specific: 256 tokens for GSM8K, HumanEval, and MBPP; 512 tokens for MATH500 and SDAR-8B-Chat; and 1024 tokens for TraDo-8B-Instruct to accommodate longer reasoning chains.

**Text-to-Image Generation.** For multimodal experiments using the MMaDa model on ImageNet and GenEval, we adopt a more conservative token temperature $\tau_{\text{token}} = 0.4$ and a matching position temperature $\tau_{\text{pos}} = 0.4$ to ensure high fidelity and alignment with text prompts. The candidate set size is kept at $N = 8$. We follow a 50-step decoding trajectory governed by a cosine schedule, which progressively reduces the number of masked positions to refine image details. For extreme acceleration tests (Section E), we utilize a 5-step linear schedule to evaluate the sampler's robustness under severe budget constraints.

**Creative Writing.** We evaluate creative writing performance using 200 prompts selected from the Alpaca dataset. Following the Min-P paper (Nguyen et al., 2024), we employ GPT-4.1 (OpenAI, 2025) as an LLM judge to assess generation quality. For this experiment, the maximum generation length is set to 1024 tokens, with a fixed block size of 16.

## C. Theoretical Analysis of the Info-Gain Objective

**Setup and notation.** At decoding step $t$, let $z_t$ denote the current state and $\mathcal{M}_t$ the set of masked positions. For each $\ell \in \mathcal{M}_t$, define the per-position predictive entropy

$$H^{(\ell)}(z_t) := H\big(X^{(\ell)} \mid z_t\big), \tag{14}$$

and the average state uncertainty

$$\mathcal{H}(z_t) := \frac{1}{|\mathcal{M}_t|} \sum_{\ell \in \mathcal{M}_t} H^{(\ell)}(z_t). \tag{15}$$

For an action $a_t$ selecting positions $A_t \subseteq \mathcal{M}_t$, the next state is $z_{t-1} = \text{Apply}(z_t, a_t)$ and $\mathcal{M}_{t-1} = \mathcal{M}_t \setminus A_t$. Following the main text, the Info-Gain utility is defined as

$$\text{IG}(a_t; z_t) := \mathcal{H}(z_t) - \mathcal{H}(z_{t-1}), \qquad C(a_t \mid z_t) := \sum_{\ell \in A_t} H^{(\ell)}(z_t), \tag{16}$$

$$J_{\text{IG}}(a_t; z_t) := \text{IG}(a_t; z_t) - C(a_t \mid z_t), \tag{17}$$

where $C(a_t \mid z_t)$ measures the immediate uncertainty of the chosen positions, and $\text{IG}(a_t; z_t)$ quantifies the reduction in overall state uncertainty.

**Expected information gain.** Sampling $a_t$ from a sampler $\pi(\cdot \mid z_t)$ induces randomness in $\text{IG}(a_t; z_t)$ and $J_{\text{IG}}(a_t; z_t)$. Let

$$\tilde{I}(A_t; z_t) := \mathbb{E}[\text{IG}(a_t; z_t)], \qquad \bar{J}_{\text{IG}}(A_t; z_t) := \mathbb{E}[J_{\text{IG}}(a_t; z_t)], \tag{18}$$

where the expectation is over the sampled token assignments while keeping $A_t$ fixed.

> **Proposition C.1** (Upper bound on expected information gain). *For any state $z_t$ and position set $A_t \subseteq \mathcal{M}_t$,*
>
> $$\tilde{I}(A_t; z_t) \leq \mathbb{E}[C(a_t \mid z_t)], \qquad \text{and thus} \quad \bar{J}_{\text{IG}}(A_t; z_t) \leq 0. \tag{19}$$

*Proof.* Define

$$\alpha := \frac{1}{|A_t|} \sum_{\ell \in A_t} H^{(\ell)}(z_t), \quad \beta := \frac{1}{|\mathcal{M}_{t-1}|} \sum_{\ell \in \mathcal{M}_{t-1}} H^{(\ell)}(z_t), \quad \gamma := \frac{|A_t|}{|\mathcal{M}_t|}. \tag{20}$$

Then $C(a_t \mid z_t) = |A_t|\alpha$ and $\mathcal{H}(z_t) = \gamma\alpha + (1 - \gamma)\beta$.

Let $Y := X^{(A_t)}$ be the sampled assignments. For any $\ell \in \mathcal{M}_{t-1}$,

$$I(Y; X^{(\ell)} \mid z_t) \leq \min(H(Y \mid z_t), H^{(\ell)}(z_t)) \leq \min(|A_t|\alpha, H^{(\ell)}(z_t)). \tag{21}$$

By the entropy-reduction decomposition,

$$\tilde{I}(A_t; z_t) = \gamma(\alpha - \beta) + \frac{1}{|\mathcal{M}_{t-1}|} \sum_{\ell \in \mathcal{M}_{t-1}} \mathbb{E}[I(Y; X^{(\ell)} \mid z_t)]. \tag{22}$$

**Case 1:** $\alpha \leq \beta$. Then $I(Y; X^{(\ell)} \mid z_t) \leq |A_t|\alpha$, giving

$$\tilde{I}(A_t; z_t) \leq \gamma(\alpha - \beta) + |A_t|\alpha \leq |A_t|\alpha = C(a_t \mid z_t). \tag{23}$$

**Case 2:** $\alpha > \beta$. Then $I(Y; X^{(\ell)} \mid z_t) \leq H^{(\ell)}(z_t)$ and

$$\tilde{I}(A_t; z_t) \leq \gamma(\alpha - \beta) + \beta \leq \alpha \leq |A_t|\alpha = C(a_t \mid z_t). \tag{24}$$

In both cases, $\tilde{I}(A_t; z_t) \leq \mathbb{E}[C(a_t \mid z_t)]$, implying

$$\bar{J}_{\text{IG}}(A_t; z_t) = \tilde{I}(A_t; z_t) - C(a_t \mid z_t) \leq 0. \tag{25}$$

$\square$

**Practical implications.** Proposition C.1 establishes that, for any fixed position set $A_t$, the expected Info-Gain utility $\bar{J}_{\text{IG}}(A_t; z_t)$ is upper bounded by zero. Equivalently, the expected commitment cost $C(a_t \mid z_t) - \text{IG}(a_t; z_t)$ is non-negative. In practice, we find that the realized $J_{\text{IG}}(a_t; z_t)$, computed along individual decoding trajectories, closely tracks this expectation. On benchmarks such as GSM8K, most observed values are near zero and only mildly negative (e.g., $J_{\text{IG}} \geq -5 \times 10^{-4}$ for more than 95% of cases), indicating that the selected actions often operate close to the theoretical upper bound.

Furthermore, $\text{IG}(a_t; z_t)$ is highly semantically sensitive: it captures not only the immediate uncertainty of the chosen positions but also how these assignments influence the uncertainty of the remaining masked positions. As a result, maximizing $J_{\text{IG}}(a_t; z_t)$ naturally discourages actions that would commit to poorly conditioned or inconsistent states, effectively preventing error propagation during decoding. This behavior manifests as a strong implicit correction mechanism, enabling the sampler to recover from suboptimal early decisions and maintain coherent, high-quality generation. Overall, the Info-Gain objective provides a computationally efficient and robust signal for action selection, balancing immediate certainty with long-term state stability throughout the iterative decoding process.

## Percentile Distribution of $J_{IG}$

*Figure 6.* Empirical distribution of $J_{IG}$ values sorted from highest to lowest. The 5th percentile is $-5 \times 10^{-4}$, indicating that the utility remains close to its theoretical upper bound in practice.

## D. Pseudocode for Info-Gain Sampler and Info-Gain Beam Search

We provide PyTorch-style pseudocode for the implementation of Info-Gain Sampler and Info-Gain Beam Search.

### D.1. Info-Gain Sampler

```python
def info_gain_sampler(model, seq_len, num_steps, N):
    # Initialize: all positions masked
    z = torch.full((1, seq_len), MASK_ID)

    # Initial forward pass
    with torch.no_grad():
        logits = model(z)   # [1, seq_len, vocab_size]

    for t in range(num_steps):
        # 1. Sample candidate actions
        candidates = action_sampler(z, model, N)

        # 2. Parallel Evaluation
        z_candidates = apply(z,candidates)
        logits_candidates = model(z_candidates)
        scores = compute_information_gain(logits,logits_candidates)

        # 3. State Transition
        best_idx = torch.argmax(scores)
        z = z_candidates[best_idx:best_idx+1]
        logits = logits_candidates[best_idx:best_idx+1]
    return z
```

*Listing 1.* Info-Gain Sampler

### D.2. Info-Gain Beam Search

```python
def ig_beam(model, seq_len, num_steps, N, beam_size):
    # Initialize the queue
```

```python
    beam = [(torch.full((1, seq_len), MASK_ID), 0.0)]

    for t in range(num_steps):
        next_beam_candidates = []
        next_f = []

        # Expand each beam
        for z, g in beam:
            # 1. Action Sampling
            candidates = action_sampler(z, model, num_candidates=N)

            # 2. Update Beam candidates
            z_candidates = apply(z, candidates)
            logits_candidates = model(z_candidates)
            transition_scores = compute_information_gain(candidates)
            state_values = compute_state(logits_candidates)

            g_candidates = g + transition_scores
            f_candidates = g_candidates + state_values

            for i in range(N):
                next_beam_candidates.append((z_candidates[i:i+1], g_candidates[i]))
                next_f.append(f_candidates[i])

        # 3. Selection
        f_tensor = torch.tensor(next_f)
        kept_indices = torch.argsort(f_tensor, descending=True)[:beam_size]
        beam = [next_beam_candidates[i] for i in kept_indices]

    beam.sort(key=lambda x: x[1], reverse=True)
    best_z, best_g = beam[0]
    return best_z
```

*Listing 2.* Info-Gain Beam Search

## E. Experiments in Extremely Low-Step Generation Scenarios

In this section, we present additional experimental results for scenarios where the number of decoding steps is severely constrained.

First, we present the results for the ImageNet-512 benchmark using the MMaDa model in a text-to-image generation setting. The experimental configuration is similar to that in Section 4.1, with the key difference being the use of a linear step schedule and a highly constrained budget of only 5 decoding steps. As shown in Figure 7, our method demonstrates significantly better visual quality and structural coherence under these extreme conditions. We also visualize the evolution of cumulative entropy on the ImageNet-512 benchmark, averaged over 10,000 sampled instances, in Figure 8.

To further evaluate the robustness of our sampler, we conduct an ultra-low step generation experiment using the Dream Model. We construct a simple writing task containing 50 prompts (e.g., "Write a story about a cat") and require the model to decode the content within a very small number of steps, for a fixed length of 80 tokens. In this extreme regime, most baselines fail to generate meaningful and coherent text. We record the "Collapse Step" for each method, defined as the minimum decoding-step budget at which fewer than 50% of the generated samples exhibit severe repetitions or grammatical errors.

For all experiments, the token sampling temperature is set to 0.4. For Info-Gain Sampler, the position sampling temperature is 0.4 and the number of candidate actions is $N = 8$. The results are summarized in Table 7.

These ultra-low-step experiments evaluate how to generate meaningful content under conditions of extremely limited information and few decoding steps. The performance of the greedy sampler is significantly inferior to that of the Info-Gain Sampler. This demonstrates that by balancing the utilization of immediate information with the information gain for future decisions, the Info-Gain Sampler produces more coherent generations within a restricted number of decoding steps.

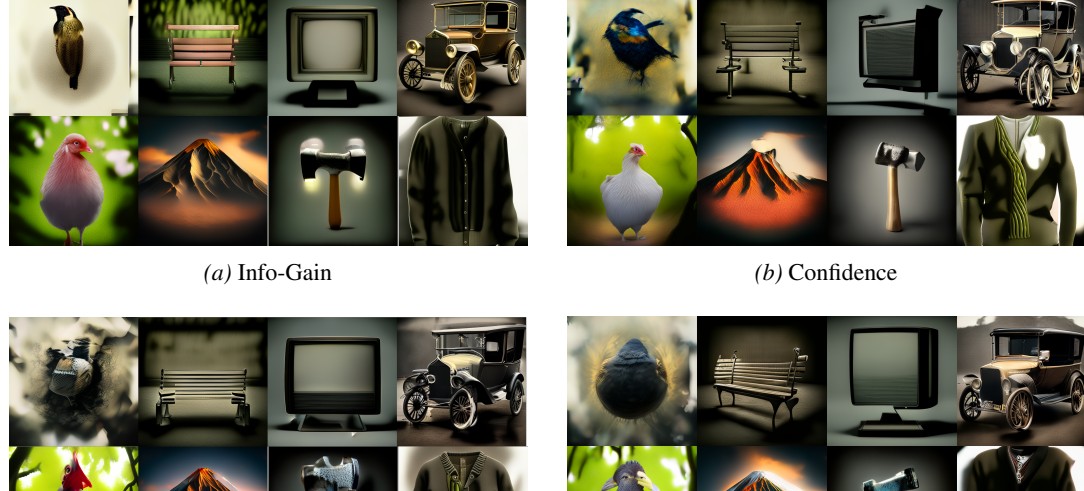

*(a)* Info-Gain

*(b)* Confidence

*(c)* Entropy

*(d)* Margin

*Figure 7.* Visual results on ImageNet-512 with an extreme budget of only 5 decoding steps. The Info-Gain Sampler maintains superior structural coherence compared to baseline heuristics.

*Table 7.* Comparison of Collapse Steps on ultra-low step writing tasks. Lower values indicate better robustness in extreme acceleration scenarios.

| Sampler | Collapse Step ($\downarrow$) |
|---|---|
| Entropy | 13 |
| Confidence | 12 |
| Margin | 12 |
| **Info-Gain** | **8** |

# F. Resource Overhead Analysis

### F.1. Memory Overhead Analysis

Let $M_W$ denote model weights, $M_A$ activations, and $M_C(L_{\text{ctx}})$ the memory of one dual-cache state for the cached context outside the current active block. Following Fast-dLLM (Wu et al., 2025b), this cached context includes both the prefix tokens before the active block and the suffix tokens after the active block; suffix positions that are not decoded in the current block remain masked but unchanged, so their KV states can also be reused. We use $N$ for the number of candidate actions evaluated per state and $B$ for the beam size. The key difference lies in whether candidate evaluation shares one outside-block dual cache or maintains multiple divergent caches:

1. **Non-cached Inference:** $M_{\text{total}} = M_W + N \cdot M_A$

2. **Info-Gain Sampler (Shared Dual Cache):** $M_{\text{total}} = M_W + M_C(L_{\text{ctx}}) + N \cdot M_A$

3. **Info-Gain Beam Search (Divergent Dual Caches):** $M_{\text{total}} = M_W + B \cdot M_C(L_{\text{ctx}}) + B \cdot N \cdot M_A$

For Info-Gain Sampler, all $N$ candidates modify only the active block while sharing the same outside-block dual cache. The additional memory cost for increasing $N$ is therefore dominated by transient activations $M_A$, which are relatively small compared to weights $M_W$. In contrast, Info-Gain Beam Search must maintain $B$ separate dual caches for divergent trajectories and evaluate $N$ candidate actions for each beam element, leading to memory that scales with both $B$ and $N$. This is shown in Table 8.

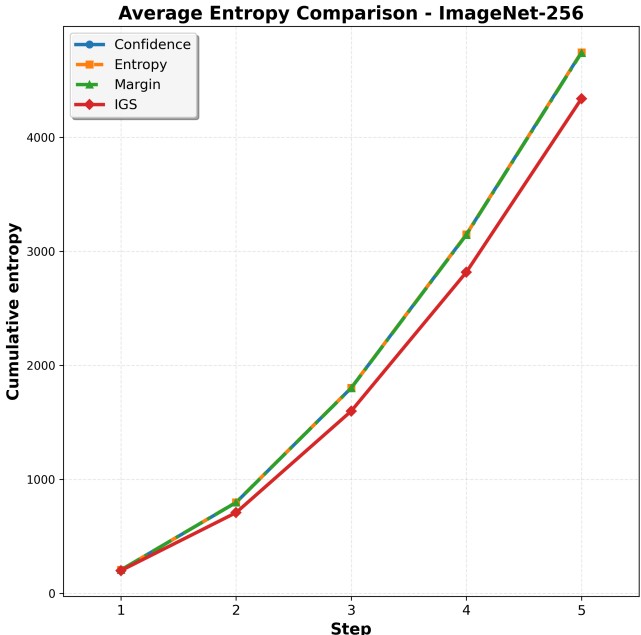

*Figure 8.* Evolution of cumulative entropy during image generation on the ImageNet-512 benchmark. The results are averaged over all labels using a 5-step linear schedule. The curves illustrate how Info-Gain Sampler manages global uncertainty compared to other methods throughout the decoding process.

*Table 8.* Memory efficiency comparison under shared dual-cache reuse.

| Method | Dual-Cache Count | Memory Scaling |
|---|---|---|
| Info-Gain Sampler | 1 shared outside-block cache | $O(M_W + M_C + N \cdot M_A)$ |
| Info-Gain Beam Search | $B$ divergent outside-block caches | $O(M_W + B \cdot M_C + B \cdot N \cdot M_A)$ |

Empirical measurements on TraDo-8B-Instruct (512 tokens, block size 16) validate this analysis. As shown in Figure 10, Info-Gain Sampler maintains stable memory usage with only 24% overhead at $N = 8$, while Info-Gain Beam Search exhibits substantial memory surge due to the $B$ divergent dual caches and the $B \cdot N$ candidate evaluations.

### F.2. Computation Overhead Analysis

The computational overhead of Info-Gain Sampler stems primarily from evaluating $N$ candidates per step. If each candidate were evaluated independently, the nominal time complexity would be:

$$T_{\text{theoretical}} = K \cdot N \cdot T_f \tag{26}$$

where $K$ is the number of decoding steps, $N$ is candidates per step, and $T_f$ is one forward pass cost.

**Practical optimizations significantly reduce overhead:**

1. **Batched Parallel Evaluation:** All $N$ candidates are processed simultaneously:

$$T_{\text{batch}} = K \cdot (T_f + \epsilon), \quad \epsilon \ll (N - 1)T_f \tag{27}$$

2. **Dual-Cache Reuse:** Reusing the shared outside-block dual cache reduces attention computation:

$$T_{\text{cached}} \approx K \cdot T_{\text{dual-cache}} + \frac{K(N - 1)}{N} T_{\text{attn}} \tag{28}$$

3. **High-confidence Bypass:** If the maximum token probability exceeds a threshold $\gamma$, the full sampling routine is skipped to reduce latency.

As shown in Fig. 9, our adaptive threshold mechanism effectively manages computational cost while maintaining high decoding quality. By dynamically adjusting the candidate acceptance threshold, we achieve near-optimal entropy reduction with minimal additional time. This explains why the practical overhead is much smaller than the naive $N\times$ estimate. While evaluating more candidates ($N$) increases per-step cost, it can reduce unnecessary full evaluations through batching, cache reuse, and high-confidence bypass. Our batched implementation and threshold optimization ensure that the quality improvements are achieved with only a modest increase in overhead.

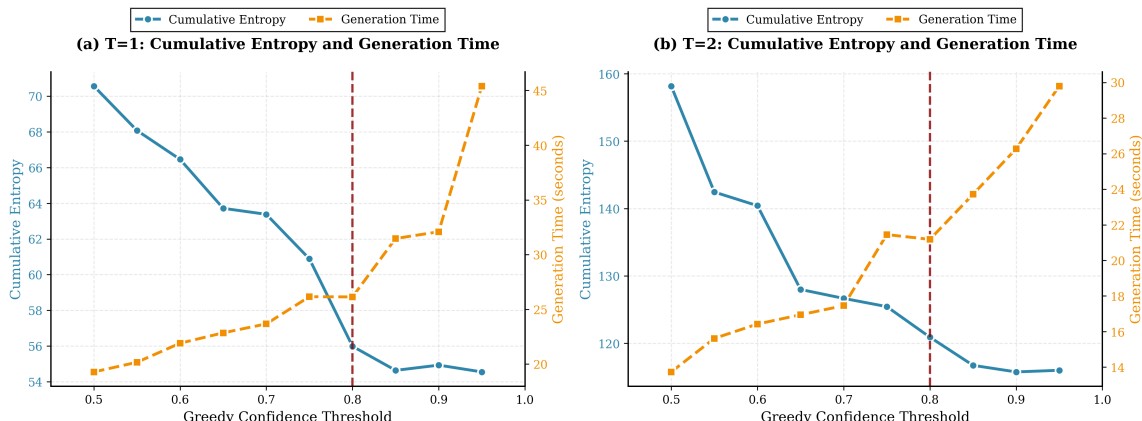

*Figure 9.* Effect of utility threshold on cumulative entropy reduction and generation time.

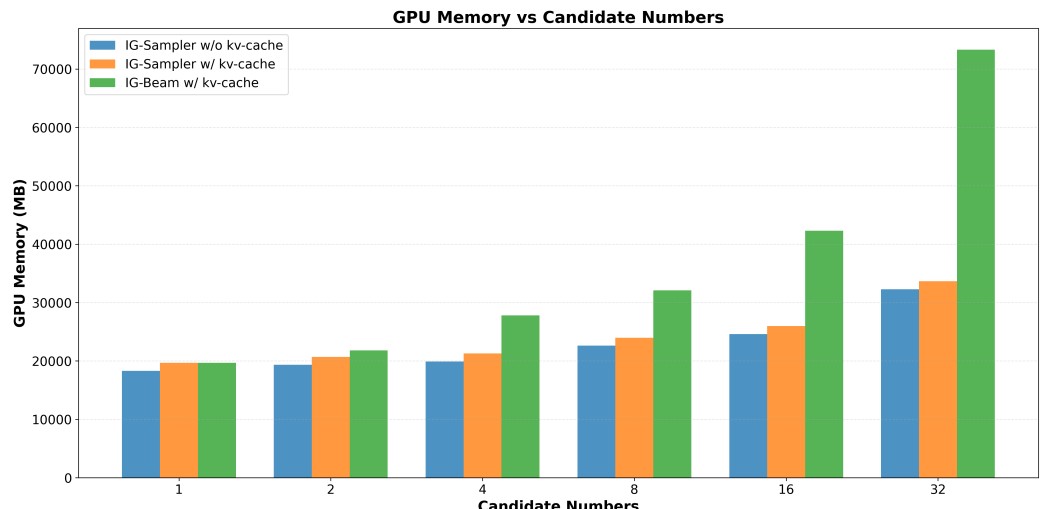

*Figure 10.* Memory usage comparison. Info-Gain Sampler maintains low overhead via shared dual-cache reuse.

