# OpenReview forum: "Improving Sampling for Masked Diffusion Models via Information Gain"
_ICML.cc/2026/Conference — ICML 2026 regular_

### Official Review · Reviewer_dRp5 · 2026-03-02

**Soundness:** 3
**Presentation:** 3
**Significance:** 3
**Originality:** 3
**Overall Recommendation:** 4
**Confidence:** 2

**Summary:**

This paper reveals that existing samplers for Masked Diffusion Models (MDMs) are typically greedily selection, which often lead to error propagation and suboptimal results. Therefore, it proposes the Information Gain Sampler, a training-free decoding framework that selects actions based on the overall uncertainty reduction. Experiments on a wide range of MDM tasks validate the effectiveness of the proposed method.

**Compliance With Llm Reviewing Policy:**

Affirmed.

**Final Justification:**

The authors' rebuttal has addressed my concerns, and I will maintain my positive score.

**Key Questions For Authors:**

1. The case study illustrates the essential of information gain, but a formal mathematical formulation could further enhance the contribution.
2. When scaling to long context, the quadratic cost of attention in the lookahead forward pass might become a bottleneck. Could the authors give some explanation?
3. Could the author clarify the core difference between this paper and related work that incorporates information gain in LLMs and diffusion models?

**Limitations:**

Yes.

**Strengths And Weaknesses:**

Strengths: The paper introduces information gain into MDMs, which transitions from a local entropy/confidence sampling strategy to a more generalized global information-theoretic perspective. It is the first to address the suboptimal performance of the greedy sampler through the lens of information gain. The experiments validate that the proposed sampler achieves consistent improvements across math, coding, planning, writing, and image generation.
Weakness: The proposed sampler needs more time and GPU consumption, which may limit its practical deployment. This paper introduces another sort of hyperparameters, which may need task-specific tuning. Thirdly, while I am not deeply familiar with this line of research,it would strength the contribution of this article to include some related work of information gain in LLMs and diffusion models.

---

> ### Author Rebuttal · Authors · 2026-03-31
>
> Response to Reviewer dRp5
>
> We sincerely thank the reviewer for the constructive comments.
>
> **[W1]** Case study formulation
>
> We thank the reviewer for the suggestion and will formalize the case study in the revision. Our corner cases illustrate a mismatch between local per-token certainty and the optimal global decoding order. For multiplication, binary product digits may appear locally more certain than the decimal factors, yet correct sequencing requires resolving factors first. For binary judgment, the answer token may appear low-uncertainty locally, but preceding reasoning steps must be completed to avoid residual errors.
>
>
> **[W2]** Lookahead forward-pass cost in long context
>
> We thank the reviewer for the question. Without KV cache, lookahead incurs quadratic attention cost and becomes a bottleneck. With KV cache (following Fast-dLLM), memory overhead is modest: for B=8 candidates and context length 32K (Trado-8B's max context length), the extra memory for parallel evaluation is ~31% of total usage, since **only per-layer activations add overhead**; the KV cache itself is shared. Wall-clock overhead under block size 4, generation length 4k and a candidate group size of 8 is ~2.9× of the single-step baseline, far below the naïve 8× cost. Detailed analysis is provided in Appendix F.
>
> **[W3]** Difference from prior information-gain methods
>
> We thank the reviewer for the question. In autoregressive LLMs, information gain has mainly been used for selection of dataset or prompt construction, such as prompt selection to improve incontext learning quality [1], and **IGPO** [2], which uses turn-level information gain as an intrinsic reward for multi-turn agent training. For acceleration, **Speculative Verification** [3] leverages information gain to control the length of speculative decoding sequences and improve verification efficiency. In diffusion language models, methods such as the **Entropy-Bounded Sampler** [4] use entropy or information-theoretic criteria to guide adaptive sampling while preserving sample quality.
>
> Unlike prior work that applies information gain mainly for training or acceleration, Info-Gain integrates it directly into inference-time decoding of masked diffusion models. At each step, it leverages the full logits to estimate the expected global uncertainty reduction over all remaining positions, enabling globally informed per-step decisions. By fully exploiting the information in each forward pass, Info-Gain improves generation quality as a principled decoding objective.
>
> [1]: *Towards Informative Few-Shot Prompt with Maximum Information Gain for In-Context Learning*. https://arxiv.org/abs/2310.08923
>
> [2]: *Information Gain-based Policy Optimization: A Simple and Effective Approach for Multi-Turn Search Agents*. https://arxiv.org/abs/2510.14967
>
> [3]: *Speculative Verification: Exploiting Information Gain to Refine Speculative Decoding*. https://arxiv.org/abs/2509.24328
>
> [4]: *Accelerated Sampling from Masked Diffusion Models via Entropy Bounded Unmasking*. https://arxiv.org/abs/2505.24857

---

> > ### Author Rebuttal · Reviewer_dRp5 · 2026-04-04
> >
> > Thank you for your detailed reply. I will maintain my positive score.

---

> > > ### Author Response · Authors · 2026-04-04
> > >
> > > We sincerely thank Reviewer dRp5 for acknowledging our rebuttal and maintaining the positive recommendation. We are glad that our responses have addressed your concerns. Should any further questions arise, we are happy to provide additional results or clarifications at any time.

---

### Official Review · Reviewer_DoGA · 2026-03-12

**Soundness:** 3
**Presentation:** 3
**Significance:** 2
**Originality:** 2
**Overall Recommendation:** 4
**Confidence:** 4

**Summary:**

Masked Diffusion Models(MDMs) have drawn extensive attentions due to its flexibility in parallel decoding paths with bidirectional attentions recently. This paper first identifies that existing greedy certainty-based samplers often fail to find optimal decoding paths and MDMs’s bidirectional architecture enables efficient information gain estimation in one forward pass. Based on these observations, this paper proposes the Info-Gain Sampler, which balances immediate certainty with long-term information gain. By exploiting MDMs’ bidirectional attention, the proposed method aims to mitigate the myopia of existing uncertainty-based heuristics that focus only on local confidence. Extensive evaluations across reasoning, coding, creative writing, and image generation, showing strong quantitative. Additionally, the theoretical analysis and efficiency considerations further enhance the contribution’s completeness.

**Compliance With Llm Reviewing Policy:**

Affirmed.

**Final Justification:**

The quantitative results and the clarifications provided during the rebuttal phase are robust and sufficiently support the paper's claims.

**Key Questions For Authors:**

please see above

**Limitations:**

yes

**Strengths And Weaknesses:**

Strengths:

1.Well-motivated design and insightful observations: the analysis on existing greedy certainty-based samplers provide insightful motivations for developing effective samplers that find optimal decoding paths.

2.Methodological soundness: The Info-Gain objective is simple yet elegant, combining immediate cost and cumulative uncertainty reduction while maintaining tractable computation through batched evaluation and KV-caching.

3.Comprehensive experiments: The evaluation spans multiple domains/tasks and architectures (full-attention, semi-autoregressive, and multimodal MDMs), demonstrating generalization and robustness. Moreover, quantitative results clearly support the claim of improved global trajectory optimization.

4.Empirical depth: Ablation analyses (entropy evolution, temperature sensitivity, beam search comparisons) strengthen confidence that the improvements arise from the proposed formulation rather than hyperparameter tuning.



Weaknesses:

1.Theoretical justification: The derivation of the Info-Gain bound is sound but lengthy, it would be better to include an intuitive, diagrammatic explanation of why balancing immediate cost and gain leads to global optimality. This could be summarized more compactly for accessibility.

2.Scalability across long sequences: The experiments on short tasks are convincing, but testing on large text-generation tasks (long output/context tokens) or long multimodal outputs would better demonstrate scalability and cumulative uncertainty performance.

3.Parameters: Clarify how temperature parameters (τ_token, τ_pos) are tuned and whether they are fixed across benchmarks. Moreover, Some baselines (e.g., KLASS, PC-Sampler) have different computational budgets or dynamic thresholds. A table clarifying per-step computational cost and decoding budget normalization would help ensure fairness.

---

> ### Author Rebuttal · Authors · 2026-03-31
>
> Response to Reviewer DoGA
>
> We thank the reviewer for the constructive feedback.
>
> **[W1]** Theoretical Justification and Intuitive Proof
>
> We will streamline the proof and add a concise intuitive summary in the revision. The key idea is that, under an information-theoretic upper bound on mutual information, the action-state mutual information in our setting can be shown to satisfy the consistency condition. Consequently, in the limit of increasing compute budget, an A*-style search guided by this heuristic recovers the optimal solution. In practice, such extreme compute is not required. Info-Gain already produces low cumulative entropy and high-quality samples under realistic budgets. Appendix Fig. 6 empirically shows that the consistency inequality holds in the vast majority of decoding steps, supporting the reliability of the heuristic in practical regimes.
>
> **[W2]** Scalability across Long Sequences and Large Outputs
>
> We analyze scalability along two key dimensions:
>
> (i)Overhead of Long Context. With KV cache, both memory and wall-clock overhead remain moderate. For example, when B=8 and the context length scales to 32K (Trado-8B maximum), the additional memory is ~31% of total usage, as the extra cost mainly comes from per-layer activations, which remain secondary compared to KV cache storage and model parameters. In terms of wall-clock, under block size 4 and generation length 4K, the overhead is ~2.9× relative to the single-step baseline, well below the naive 8× factor.
>
> (ii)Performance. The main limitation of long-context generation lies in the base model capability rather than the sampling strategy. On S-NIAH-16k in RULER with Trado-8B, all methods (including ours) fail to produce meaningful outputs, indicating that the predictive distribution becomes unreliable in this regime. Under such conditions, although Info-Gain continues to reduce cumulative entropy (e.g., ~0.6× of the confidence baseline), this does not necessarily translate into downstream performance gains. In moderate-length generation (e.g., length 1K), where the model predictions remain informative, Info-Gain consistently outperforms other baselines on creative writing tasks across temperatures T $\in$ {0.5, 1.0, 1.5}, supporting that its effectiveness depends on the quality of the underlying predictive distribution.
>
> **[W3-1]** Parameters and Hyperparameter Settings
>
> We do not tune temperatures per task or per baseline, and instead adopt fixed settings aligned with common practice. For reasoning tasks, certainty-greedy baselines use τ_token = 0.7 and τ_pos = 0, while Info-Gain uses τ_token = 0.7 and τ_pos = 0.1 to retain mild stochasticity in action selection. For MMaDa T2I, we set τ_token = τ_pos = 0.4. Sensitivity analyses are provided in Sec. 4.3 and Fig. 5. For other baselines, PC-Sampler uses α = 10 and λ = 0.25. For KLASS, we set `kl_threshold = 1e-4`. Dynamic thresholding is disabled to ensure fairness under a shared NFE budget.
>
> **[W3-2] Computational Budget and Fair Comparison**
>
> We ensure fairness by explicitly controlling the number of decoded tokens per step. Info-Gain differs only at steps involving candidate branching, while strictly maintaining the same overall decoding budget. This ensures a strictly matched computational budget across all methods. With our optimized implementation (dual KV cache + high-confidence bypass), the end-to-end wall-clock remains within 2× of baselines such as PC-Sampler (Appendix F, Table 9). **To further validate fairness**, we additionally compare three classes of methods under the same budget (B=8): (i) aggregation-based methods, including Majority Voting@8 (sampling 8 trajectories and voting on parsed answers) and Min-Cumulative-Entropy@8 (selecting the sample with the lowest cumulative entropy), both built on PC-Sampler; (ii) a lookahead-based method, LookUM (B=8, entropy as the scoring function); and (iii) our sampling-based method, Info-Gain.
>
> | Task  | PC Majority@8 | PC Min-CE@8 | LookUM| Info-Gain (Ours) |
> |-------|-------|--------|--------|-----------|
> | GSM8K| 86.1|82.1|78.3|83.3|
> | MATH500| 54.0|49.2|51.4|51.3|
> | HumanEval|–|51.8|52.3|59.2|
> | MBPP|–| 44.9|45.9|48.4|
> | Sudoku|–|81.3|78.4|84.4|
> | Countdown|–|41.9|43.3|45.2|
>
> Info-Gain achieves consistently strong performance across tasks, outperforming other compute-matched baselines on precision-sensitive tasks (coding and planning), while remaining competitive on math. In settings with a small number of distinct, easily extractable answers (e.g., GSM8K), Majority Voting@8 can be slightly stronger due to answer-level aggregation. As a sampler, Info-Gain is more memory-efficient than aggregation-based approaches, since it enables KV cache sharing within a single decoding process, whereas aggregation requires multiple independent trajectories without shared prefix states. In addition, Info-Gain is more generally applicable, as it does not rely on answer-level comparability and can be directly applied to structured generation tasks.

---

> > ### Author Rebuttal · Reviewer_DoGA · 2026-04-02
> >
> > Thank you for the detailed rebuttal. I will maintain the positive score recommendation.

---

> > > ### Author Response · Authors · 2026-04-03
> > >
> > > We sincerely thank Reviewer DoGA for acknowledging our rebuttal and maintaining the positive recommendation. We are glad that our responses have addressed your concerns. Should any further questions arise, we are happy to provide additional results or clarifications at any time.

---

### Official Review · Reviewer_Mdsi · 2026-03-12

**Soundness:** 3
**Presentation:** 2
**Significance:** 2
**Originality:** 3
**Overall Recommendation:** 4
**Confidence:** 3

**Summary:**

This paper argues the typical MDM sampling where tokens are predicted and then remasked based on uncertainty is too myopic, and that the locally greedy remasking can be improved by accounting for the information gain across timesteps. They propose a new sampling algorithm for MDM that does a better job minimizing the cumulative entropy of the trajectories.

**Compliance With Llm Reviewing Policy:**

Affirmed.

**Final Justification:**

The results in the paper and rebuttal are convincing. I still have concerns about the writing quality.

**Key Questions For Authors:**

(Q1) I’m curious what the entropies of the marginals are throughout sampling. What does the line plot entropy x timestep look like? I suspect that discrete diffusion has low entropy initially and then very high entropy at the end after it’s backed itself into a corner and then the entropy explodes, is this the case?

(Q2) What does “near-optimal” in Observation 1 mean?

(Q3) How does your method perform in the setting from Table 1?

**Limitations:**

Yes

**Strengths And Weaknesses:**

Strengths

(S1) Intuitive motivation and story.

(S2) Consistent performance improvement over the benchmarked samplers.

(S3) Nice analysis of cumulative entropy vs. accuracy.


Weaknesses

(W1) The writing needs significant revision for clarity, especially regarding equations. It is important to explicitly define and explain notation and equations. For example H^{(l)} is used on line 152/153 is extremely confusing to parse. From context I believe H is entropy so H^{(l)} this is plogp at position l of z_t, however, z_t is defined to be a string of tokens in line 095 and not a distribution. Ignoring this inconsistency, the notation is still confusing because l doesn’t really index H it indexes z_t.

(W2) The argument regarding “MDM don’t have the next-token bottleneck” seems tenuous, since one can argue this significantly increases the search space requiring picking both the correct token and position as opposed to just the next token.

(W3) Even for MDM, autoregressive sampling tends to be a strong baseline and should be reported in all settings.

(W4) Inconsistent hyphenation. Uses both autoregressive (e.g. line 101) and auto-regressive (e.g. line 142).

(W5) There needs to be some wall clock comparison of the samplers, especially because one of the listed core contributions is an efficient implementation of Info-Gain sampler.

---

> ### Author Rebuttal · Authors · 2026-03-31
>
> Response to Reviewer Mdsi
>
> We sincerely thank the reviewer for the meticulous reading and the insightful questions.
>
> **[W1&W4]** Clarity of Writing and Notation
>
> We agree that $H^{(l)}(z_t)$ was unclear. We will consistently update all notations and unify “auto-regressive” throughout the manuscript. Following submission, we have spent the past two weeks carefully refining the writing, and these improvements will be reflected in the revised version.
>
> **[W2]** The "Next-token Bottleneck" in MDM vs. AR
>
> We agree that parallel decoding in MDMs can introduce errors due to dependencies in the joint distribution ( $q(a,b|context)\neq p(a|context)p(b|context)$ ), as discussed in some papers like *Generation Order and Parallel Decoding in Masked Diffusion Models*. However, our claim concerns the evaluation of any decoding actions, instead of the real sampling process.MDMs provide simultaneous access to marginal distributions at all positions, enabling efficient global uncertainty evaluation. In contrast, AR models can only perform such evaluation sequentially, via token-by-token generation.
>
> **[W3]** Lack of AR Baseline
>
> As suggested, we include AR baselines. Compared to standard baselines such as confidence, AR performs competitively on math tasks, significantly worse on coding and planning, and similar or slightly better on creative writing tasks.
>
>
>
> Table 1: Reasoning Tasks Performance. K means the number of token to be decoded at each step.
>
> | method | GSM8K | MATH500 | HumanEval | MBPP | Sudoku | Countdown | $\tilde{H}$ |
> | :--- | :---: | :---: | :---: | :---: | :---: | :---: | :---: |
> | **AR (K=1)** | 76.5% | 43.8% | 41.9% | 35.6% | 61.8% | 35.5% | 121.7 |
> | **AR (K=2)** | 53.8% | 23.0% | 23.7% | 22.2% | 38.2% | 25.8% | 273.1 |
>
> Table 2: Creative writing win-rate (%) of Info-Gain vs. AR
>
> | Temp | K | Win-rate vs. AR |
> | :---: | :---: | :---: |
> | 0.5 | 1 | 60.1% |
> | 0.5 | 2 | 63.9% |
> | 1.0 | 1 | 54.8% |
> | 1.0 | 2 | 65.2% |
> | 1.5 | 1 | 58.4% |
> | 1.5 | 2 | 69.7% |
>
> **[W5]** Wall-Clock Comparison and Efficiency
>
> We thank the reviewer for the comment. End-to-end wall-clock comparisons are reported in Appendix F (Table 9). After our efficient implementation, wall-clock time is reduced to **less than 2×** of the one token per step baseline such as confidence or PC-Sampler. The main speedups come from two sources:
>
> (i) KV Cache enables efficient parallel evaluation
> KV cache, as used in fast-DLLM, drastically reduces per-step computation. For Dream-7B with prefill length 128 and generation length 512 with KV cache, evaluating 8 candidates each step takes 45.4s, compared to 16.3s for B=1 each step, only ~3× slower. The additional memory cost of batching is acceptable for current diffusion LMs(Trado-8B,SDAR-8B, etc.), which support up to 32k sequence length.
>
> (ii) High-Confidence Bypass skips trivial choices
> At each step, if any token in the current block exceeds a confidence threshold (0.8 in our experiments), Info-Gain computation is skipped, the high confidence token will be directly selected. About 66% of steps bypass IG computation, significantly reducing overhead without noticeably affecting sample quality (see Appendix Fig. 9 for the trade-off with cumulative entropy).
>
>
> **[Q1]** Entropy vs. Timestep Patterns
>
> In our observations, average marginal entropy follows two regimes. On correct trajectories it decreases steadily. On erroneous trajectories it often falls at first as well, but after a serious mistake it can spike sharply, reflecting high uncertainty in the resulting state, and then declines again toward the end of decoding. The Info-Gain objective aligns well with this pattern, because such spikes mark steps where reducing uncertainty is especially informative. Visualizations and further details: [anonymous.4open.science](https://anonymous.4open.science/r/info-gain-0001/).
>
> **[Q2]** Clarification of "Near-optimal" in Observation 1
>
> In our context, “Near-optimal” refers to whether a greedy decoding trajectory is close to the optimal one under cumulative entropy, or can deviate significantly from it. We use "near-optimal" because under some tasks(e.g., submodular-like objectives such as set cover), a greedy strategy can be close to or exactly the optimal one. Observation 1 shows that such greedy strategies can incur large deviations in our proposed two-way corner cases.
>
> **[Q3]** Performance in the Table 1 “Judge” Setting
>
> In the judge-style setting of Table 1, Info-Gain reaches cumulative entropy 24.87 nats, better than AR and the other baselines, while its accuracy is 87%, below AR but above the remaining baselines. This pattern is expected: Info-Gain explicitly optimizes position and token jointly under a cumulative-entropy objective, so it pushes uncertainty down aggressively. For this binary judge task, AR order is near-oracle, so it slightly outperforms in raw accuracy despite higher cumulative entropy.

---

> > ### Author Rebuttal · Reviewer_Mdsi · 2026-04-03
> >
> > I sincerely appreciate the authors' effort during the rebuttal. The AR decoding results added during rebuttal are compelling and address (W3). The wall-clock analysis I missed in the appendix addresses (W5).
> >
> > My writing concerns remain (W1, W2, W4), but in light of the results I will increase my score and trust the authors will take the necessary time to revise the manuscript in the interest of clarity.

---

> > > ### Author Response · Authors · 2026-04-04
> > >
> > > We sincerely thank the Reviewer Mdsi for acknowledging our rebuttal and the reasonable discussion. We are glad that our responses have addressed W3, W5, and Q1–3.
> > >
> > > Regarding W2 (Next-token bottleneck in MDM): We would like to further elaborate as follows:
> > >
> > > - **Larger search space theoretically enables MDM to find higher-quality samples.** The bottleneck you mentioned would be better termed the **Path-Selection Bottleneck**. Importantly, this does not prevent MDM from decoding in a fixed, simple manner (e.g., position-wise AR or highest-confidence-first), making it at least as powerful as AR in terms of path selection. Theoretically, as long as MDM decodes only one token per step (like AR), it can still achieve lossless modeling without theoretical approximation error. How to increase parallelism without introducing such error is highly dependent on data structure.
> > >
> > > - **Our definition of the "next-token bottleneck"** refers to how many positions' distributional information is available at a single instant. MDM can obtain marginal distributions for *all* positions simultaneously, whereas AR can only access the next position's distribution. This is the most direct meaning of the bottleneck we intend to highlight.
> > >
> > > - **Our method fully leverages global forward information** to identify the optimal path, thereby raising the upper bound of MDM within its vast search space. From this perspective, the absence of a Next-token bottleneck in MDM can compensate for the path-selection bottleneck you pointed out during sampling.
> > >
> > > Thank you for your constructive feedback. Should any further questions arise, we are happy to provide additional results or clarifications at any time.

---

### Official Review · Reviewer_6B2C · 2026-03-13

**Soundness:** 3
**Presentation:** 3
**Significance:** 3
**Originality:** 3
**Overall Recommendation:** 4
**Confidence:** 3

**Summary:**

The paper presents a training-free sampling approach for masked diffusion models. The premise is straightforward: current sampling approaches tend to greedily optimize positions to unmask with lowest certainty and do not account for future cumulative uncertainties of remaining masked positions, leading to suboptimal unmasking actions. To address this, authors propose “Info-Gain” based sampler where the unmasking action is determined on both uncertainty of current step unmasking candidates and the ‘information gain’ based on unmasking candidates (i.e. the reduction in state uncertainty / average marginal entropy over remaining masked candidates). The method shows beneficial results on multiple architectures and tasks including reasoning, coding, image generation and creative writing, and outperforms existing samplers for masked diffusion models. Authors also conduct ablations on beam search, best of n sampling, etc and robustness to sampling temperature settings.

**Compliance With Llm Reviewing Policy:**

Affirmed.

**Key Questions For Authors:**

Please see weaknesses

**Limitations:**

yes

**Strengths And Weaknesses:**

Strengths:


- Method is straightforward and relatively novel in utilizing look-ahead uncertainty estimation to perform sampling based on both current uncertainty and future uncertainty based on selected unmasking positions.
- Empirical results show benefits on multiple benchmarks and tasks including coding, reasoning, image generation and creative writing.
Ablations are adequately conducted to show impact of expansion budget and beam size, temperature sensitivity, analysis of cumulative entropy etc


Weaknesses:

- Lack of compute-matched baseline evaluation: The method requires additional compute (with a group size N=8) leading to more generation time and GPU memory usage than existing samplers; however they do not provide a compute-matched comparison with existing samplers (e.g. confidence-based sampling with a lookahead of 2 steps and beam size of 8 should be considered for compute-matched comparison).
- Impact of lookahead steps and group size is missing: Currently the method only accounts for 2 step lookahead and group size of 8, but tradeoff between group size and lookahead steps is missing (e.g. given a budget of B, would it be better to sample action impact over more steps and how does the group size impact. These are important ablations to consider

---

> ### Author Rebuttal · Authors · 2026-03-31
>
> Response to Reviewer 6B2C
>
> We sincerely thank the reviewer for the constructive comments.
>
> **[W1]** Lack of compute-matched baseline evaluation
>
> We thank the reviewer for raising fairness under equal computational budget. To provide a fairer comparison, we evaluated three families of methods under the same computational budget on Dream-7B across Mathematics (GSM8K, MATH500), Coding (HumanEval, MBPP), and Planning (Sudoku, Countdown).
>
> 1. Aggregation: `Majority Voting@8` (sampling 8 samples and voting on parsed answers) and `Min-Cumulative-Entropy@8` (selecting the single sample with the lowest cumulative entropy across 8 samples). These baselines were implemented on top of the strongest baseline, the PC-sampler.
> 2. Lookahead: `LookUM` ($B=8$, entropy as score function), a concurrent lookahead-based method.
>
> Table 1: Performance comparison under the same computational budget.
>
> | Task | PC `Majority@8` | PC `Min-Cum-Entropy@8` | LookUM (B=8, entropy) | Info-Gain (Ours, B=8) |
> | :--- | :---: | :---: | :---: | :---: |
> | GSM8K | 86.1 | 82.1 | 78.3 | 83.3 |
> | MATH500 | 54.0 | 49.2 | 51.4 | 51.3 |
> | HumanEval | \ | 51.8 | 52.3 | 59.2 |
> | MBPP | \ | 44.9 | 45.9| 48.4 |
> | Sudoku | \ | 81.3| 78.4 | 84.4 |
> | Countdown | \ | 41.9 | 43.3 | 45.2 |
>
> Analysis:
> - Majority Voting: Strong on tasks with deterministic answer extraction (e.g., GSM8K), even surpassing Info-Gain. Not applicable to Coding/Planning. Parallel paths cannot share the same prefix KV cache; unbatched evaluation is slower than Info-Gain. It would also be possible to integrate majority voting into Info-Gain, but it is not the focus of this paper.
> - Min-Cumulative-Entropy: Universal but consistently underperforms Info-Gain across tasks.
> - LookUM: Comparable on Math, but Info-Gain wins clearly on Code/Planning (+6.9pp HumanEval, +2.5pp MBPP, +6.0pp Sudoku, +1.9pp Countdown). Math gains are smaller (+5.0pp GSM8K; MATH500 similar). This means the objective of Info-Gain is more suitable for precision-sensitive tasks.
>
> Info-Gain provides superior performance on precision-sensitive tasks (6.9% better on HumanEval and 3.1% better on Sudoku), while remaining competitive on math tasks, validating the effectiveness of our per-step information gain objective under equal computational budget.
>
>
> **[W2]** Impact of lookahead steps and group size is missing
> We thank the reviewer for the suggestion to ablate how lookahead budget is split between action group size $N$ and lookahead depth $L$. We added an ablation with fixed budget $B = N \times L = 8$, using greedy-confidence selection for the lookahead. For multi-step lookahead ($L > 1$), the objective is extended to the sum of multi-step $J_{IG}$.
>
> Table 2: Ablation of action group size ($N$) vs. lookahead depth ($L$) on MATH500 and Countdown.
>
> | Task | N | L | Accuracy | Cum-Entropy | Avg. Time (s/q) |
> | :---: | :---: | :---: | :---: | :---: | :---: |
> | MATH500 | 8 | 1 | 51.3 | 70.96 | 45.44 |
> | | 4 | 2 | 50.1 | 99.01 | 54.57 |
> | | 2 | 4 | 48.2 | 86.20 | 76.90 |
> | Countdown | 8 | 1 | 45.2 | 10.84 | 3.48 |
> | | 4 | 2 | 46.8 | 12.75 | 5.31 |
> | | 2 | 4 | 45.8 | 13.91 | 8.05 |
>
> Analysis:
> - Task dependency: The budget allocation is not universally optimal but task-dependent. On MATH500, $(N,L)=(8,1)$ achieves the best accuracy, so shallower, wider lookahead is preferable. On Countdown, $(4,2)$ outperforms both $(8,1)$ and $(2,4)$. This aligns with planning-style tasks where the number of plausible actions per state is relatively small, allocating some budget to multi-step lookahead can improve action choice.
> - Efficiency and stability: $(8,1)$ gives the lowest cumulative entropy in this ablation on both tasks. With $L=1$, the lookahead evaluation can be done in a single batched forward pass, which is the most time-efficient and practical option. When $L>1$, the lookahead evaluation can't be done in a single forward pass.
>
> For efficiency, simplicity, and stable wall-clock across tasks, we use 1-step lookahead as the default.

---

> > ### Author Rebuttal · Reviewer_6B2C · 2026-04-03
> >
> > Thanks for the rebuttal, i will maintain my score.

---

> > > ### Author Response · Authors · 2026-04-04
> > >
> > > We sincerely thank Reviewer 6B2C for acknowledging our rebuttal and maintaining the positive recommendation. We are glad that our responses have addressed your concerns. Should any further questions arise, we are happy to provide additional results or clarifications at any time.

---

### Decision · Program_Chairs · 2026-04-30

**Decision:**

Accept (regular)

**Comment:**

This paper proposes the Info-Gain Sampler, a training-free decoding framework for masked diffusion models that balances immediate cost with long-term information gain to reduce cumulative uncertainty. The reviewers agreed that transitioning from locally greedy sampling to a global, information-theoretic decoding is conceptually novel and found the consistent empirical improvements across diverse tasks to be compelling. However, concerns were initially raised regarding the lack of compute-matched baselines, potential wall-clock overhead, and long-context scalability. In the rebuttal, the authors provided compute-matched comparisons, shared detailed wall-clock measurements enabled by their KV-caching and high-confidence bypass, and clarified notation inconsistencies. These responses were well-received: Reviewer Mdsi raised their score from a reject to a weak accept, and the other three reviewers (6B2C, DoGA, dRp5) maintained their positive ratings. Given that all reviewers recommend acceptance and their technical concerns have been resolved, the submission is recommended for acceptance.